# Zeroth-Order Optimization Finds Flat Minima

**Liang Zhang**
ETH Zurich & Max Planck Institute
liang.zhang@inf.ethz.ch

**Bingcong Li**
ETH Zurich
bingcong.li@inf.ethz.ch

**Kiran Koshy Thekumparampil**
Amazon
kkt@amazon.com

**Sewoong Oh**
University of Washington
sewoong@cs.washington.edu

**Michael Muehlebach**
Max Planck Institute
michaelm@tuebingen.mpg.de

**Niao He**
ETH Zurich
niao.he@inf.ethz.ch

## Abstract

Zeroth-order methods are extensively used in machine learning applications where gradients are infeasible or expensive to compute, such as black-box attacks, reinforcement learning, and language model fine-tuning. Existing optimization theory focuses on convergence to an arbitrary stationary point, but less is known on the implicit regularization that provides a fine-grained characterization on which particular solutions are finally reached. We show that zeroth-order optimization with the standard two-point estimator favors solutions with small trace of Hessian, which is widely used in previous work to distinguish between sharp and flat minima. We further provide convergence rates of zeroth-order optimization to approximate flat minima for convex and sufficiently smooth functions, where flat minima are defined as the minimizers that achieve the smallest trace of Hessian among all optimal solutions. Experiments on binary classification tasks with convex losses and language model fine-tuning support our theoretical findings.

## 1 Introduction

There are many emerging machine learning problems where gradients are not accessible or expensive to compute, hindering the application of gradient-based optimization algorithms. For example, fine-tuning large language models (LLMs), particularly at the scale of billions of parameters, faces significant memory bottlenecks, primarily because of the memory-intensive nature of backpropagation. Zeroth-order optimization offers a compelling alternative as it permits gradient estimation via finite differences of loss values. Malladi et al. [69] reported that zeroth-order methods are capable of fine-tuning a 30-billion-parameter model using a single A100 GPU with 80 GiB memory, whereas gradient-based methods require 8 A100s. In addition to recent advances in fine-tuning LLMs, zeroth-order methods have also found numerous applications in black-box settings [17, 18, 6] and nonsmooth optimization [74, 57, 47] where gradient computation is often infeasible. They have further proven effective in reinforcement learning [80, 68, 39] and distributed learning [31, 93, 77] to reduce computation and communication costs.

To be more specific, for the optimization problem $\min_{x \in \mathbb{R}^d} f(x)$ with parameters $x \in \mathbb{R}^d$ and a loss function $f : \mathbb{R}^d \to \mathbb{R}$, zeroth-order optimization with the standard two-point gradient estimator [74]

39th Conference on Neural Information Processing Systems (NeurIPS 2025).

(Algorithm 1) iteratively updates $x$ by substituting the computationally intractable gradient with

$$g_\lambda(x, u) \quad := \quad \frac{f(x + \lambda u) - f(x - \lambda u)}{2\lambda} u, \tag{1}$$

where $u \sim \mathcal{N}(0, \mathrm{I}_d)$ is a standard Gaussian random vector and $\lambda \in \mathbb{R}$ is a smoothing parameter. It is convenient to understand zeroth-order methods through a surrogate smoothed function defined as $f_\lambda(x) := \mathbb{E}_{u \sim \mathcal{N}(0, \mathrm{I}_d)}[f(x + \lambda u)]$ [27, 74]. Since Eq. (1) unbiasedly estimates the gradient of $f_\lambda(x)$ [74], i.e., $\mathbb{E}[g_\lambda(x, u)] = \nabla f_\lambda(x)$, standard convergence analyses directly indicate that $f_\lambda(x)$ is minimized. Further noting that $f_\lambda(x)$ is close to $f(x)$ when $\lambda$ is small, the convergence of zeroth-order optimization can be established on the original loss $f(x)$. For example, when $f(x)$ is smooth and convex, zeroth-order methods guarantee that the average of the iterates, $\bar{x}_T$, after $T$ iterations satisfy $\mathbb{E}[f(\bar{x}_T) - \min_{x \in \mathbb{R}^d} f(x)] \leq \mathcal{O}(d/T)$; see e.g., [74].

However, in the presence of multiple solutions, it remains unclear from the above arguments whether zeroth-order methods prefer certain minima. Our intuition to this question comes from revisiting the underexplored role of $f_\lambda(x)$. Using Taylor's theorem (with details in Eq. (2)), one can find that

$$f_\lambda(x) \quad = \quad f(x) + \frac{\lambda^2}{2} \mathrm{Tr}\left(\nabla^2 f(x)\right) + o(\lambda^2).$$

This implies that zeroth-order optimization implicitly encodes an additive regularizer using the trace of Hessian, which is a widely adopted metric in the literature [89, 88, 58, 34, 24, 3] to differentiate sharp and flat minima [40, 46, 29, 75, 25, 38]. Existing works studying flat minima mostly centered around first-order methods. The work of [75, 105, 85] empirically demonstrated that stochastic gradient descent (SGD) converges to solutions with small expected sharpness $\mathbb{E}_{u \sim \mathcal{N}(0, \mathrm{I}_d)}[f(x + \lambda u)] - f(x)$ on a variety of vision tasks, which also implies small trace of Hessian according to Eq. (2). It was also shown that SGD with label noise provably decreases trace of Hessian as a regularization term for overparameterized models [12, 22, 55]. Wen et al. [89] proved that sharpness-aware minimization (SAM) [33], a method specifically designed for finding flat minima, minimizes trace of Hessian when the batch size is one. Further discussions on the relevance of flat minima, as well as the role of the trace of Hessian can be found in the recent work [3].

This work formalizes the intuition above and initiates the study of the implicit regularization of zeroth-order optimization with the standard two-point estimator. Despite relying only on function evaluations of $f(x)$, we show that zeroth-order optimization converges to flat minima, which are typically characterized using second-order information such as the Hessian matrix $\nabla^2 f(x)$. In particular, our contributions are summarized below.

• We define flat minima as the minimizers that achieve the smallest trace of Hessian over the set of all minimizers; see Definition 3.1. Assuming the function is convex and three times continuously differentiable with Lipschitz-continuous gradient, Hessian, and third derivatives (Assumptions 3.3 and 3.4), we prove that zeroth-order optimization with the standard two-point estimator (Algorithm 1) converges to $(\mathcal{O}(\epsilon/d^2), \epsilon)$-approximate flat minima, defined in Definition 3.2, after $T = \mathcal{O}(d^4/\epsilon^2)$ iterations; see Corollary 2. While standard convergence analysis directly treats $\lambda^2 \mathrm{Tr}(\nabla^2 f(x))$ as a bias term and controls it by choosing a sufficiently small $\lambda$, we provide a tighter and novel characterization in Section 3.1 of zeroth-order update dynamics to analyze convergence of $F(x) := f(x) + (\lambda^2/2)\mathrm{Tr}(\nabla^2 f(x))$. This result is of independent and broader interest for advancing the understanding of zeroth-order optimization and naturally extends to analyzing convergence rates of first-order methods such as SAM and SGD on the smoothed loss towards flat minima (Remark 3.4).

• We provide empirical evaluations to examine the behavior of the trace of Hessian under zeroth-order optimization across three settings: a test function (Figure 1), binary classification tasks using overparameterized SVMs and logistic regression (Figure 2), and language model fine-tuning tasks with RoBERTa [61] (Figure 3). Consistent with our theoretical predictions, we observe that the trace of Hessian decreases when using zeroth-order optimization across all these settings. Note that we adopt an estimation of the trace of Hessian on language models for scalability purposes.

To the best of our knowledge, this is the first work to prove that zeroth-order optimization converges to flat minima. Note that all results provided in this paper can be readily extended to zeroth-order optimization with other unbiased estimators of $\nabla f_\lambda(x)$ satisfying $\mathbb{E}[uu^\top] = \mathrm{I}_d$ such that Eq. (2) holds, including the one-point estimator suggested in [32, 74], as well as gradient estimation using random vectors uniformly distributed on the Euclidean sphere [81, 99].

**Algorithm 1** Zeroth-Order Optimization with the Two-Point Estimator

---

**Input:** Initialization $x_0 \in \mathbb{R}^d$, number of iterations $T$, stepsize $\eta > 0$, smoothing parameter $\lambda > 0$.

1: **for** $t = 0, 1, \cdots, T - 1$ **do**
2:     Sample $u_t$ uniformly from the standard multivariate Gaussian distribution $\mathcal{N}(0, \mathrm{I}_d)$.
3:     Construct the two-point gradient estimator

$$g_\lambda(x_t, u_t) = \frac{f(x_t + \lambda u_t) - f(x_t - \lambda u_t)}{2\lambda} u_t.$$

4:     Update the parameter

$$x_{t+1} \;\leftarrow\; x_t - \eta\, g_\lambda(x_t, u_t).$$

**Output:** $x_\tau$ for $\tau$ sampled uniformly at random from $\{0, 1, \cdots, T - 1\}$.

---

**Notation.** We use $\|\cdot\|$ for the Euclidean norm of vectors and $[m]$ for the set $\{1, 2, \cdots, m\}$. The trace of a square matrix $J$ is denoted by $\mathrm{Tr}(J)$. A standard Gaussian random vector in $\mathbb{R}^d$ is written as $v \sim \mathcal{N}(0, \mathrm{I}_d)$ and satisfies $\mathbb{E}[vv^\top] = \mathrm{I}_d$. A function $p : \mathbb{R}^d \to \mathbb{R}$ is convex if $\alpha\, p(x) + (1 - \alpha)\, p(y) \geq p(\alpha x + (1 - \alpha)y), \forall x, y \in \mathbb{R}^d$ and $\alpha \in (0, 1)$. A function $q : \mathbb{R}^d \to \mathbb{R}$ is $r$th-order smooth with $L_r > 0$ if it is $r$ times differentiable and $\|\nabla^r q(x) - \nabla^r q(y)\| \leq L_r \|x - y\|$, where $\|\nabla^r q(x)\| := \max_{s \in \mathbb{R}^d, \|s\| \leq 1} |\nabla^r q(x)[s]^r|$ and $\nabla^r q(x)[s]^r := \sum_{i_1, \cdots, i_r \in [d]} (\partial^r q(x)/\partial x_{i_1} \cdots \partial x_{i_r}) s_{i_1} \cdots s_{i_r}$. We say for brevity that $q(x)$ is $L_1$-smooth if $r = 1$.

## 1.1 Related Works

**Zeroth-Order Optimization.** Existing works primarily focused on convergence to a stationary point, while we prove the first result on convergence to flat minima. The development and early advances of zeroth-order optimization can be found in [70, 21]. Nesterov and Spokoiny [74] provided a convergence analysis of zeroth-order optimization across various settings. Their results for nonsmooth convex functions were refined by [81], while improvements for nonsmooth nonconvex functions were made by [57, 16, 47]. Extensions to the stochastic setting were considered in [36]. Lower bounds were also provided in [90, 28], showing that the dimension dependence in the convergence guarantees of zeroth-order optimization is unavoidable without additional assumptions. Several recent works [96, 69, 99] proved that such dimension dependence can be relaxed to a quantity related to the trace of Hessian. Zeroth-order optimization has been extended to minimax optimization [86], bilevel optimization [1], constrained optimization [10, 64], and Riemannian optimization [52, 53]. It has also been integrated with coordinate descent [56], conditional gradient descent [9], SignSGD [60], and variance reduction techniques [59, 30, 43]. A line of work [83, 66, 10, 98, 79] established convergence to second-order stationary points, demonstrating that zeroth-order methods can also escape saddle points. The noisy function evaluation setting was studied in [8, 65, 4], where higher-order smoothness assumptions were used to reduce bias in gradient estimates. The capability of zeroth-order methods for fine-tuning LLMs was first demonstrated by Malladi et al. [69]. Following this work, several recent studies have introduced various improvements aimed at enhancing runtime efficiency and performance, including momentum [102, 44], variance reduction [35], sparsification [37, 63], use of Hessian information [103], and better sampling strategies [19].

**Sharpness-Aware Minimization.** Prior studies of flat minima have mostly centered on first-order methods. Algorithms designed to find flat minima have achieved strong empirical success, including Entropy-SGD [15], stochastic weight averaging [42, 45], and sharpness-aware minimization (SAM) [33]. Similar methods to SAM were proposed in [92, 104], and their efficiency and performance were further enhanced in e.g., [49, 62, 26, 106, 5, 50]. Also inspired by the insights in Eq. (2), Zhang et al. [97] proposed a gradient-based method that effectively minimizes the smoothed loss. Several recent studies [95, 78, 94] proposed modifying zeroth-order optimization and combining it with principles from SAM to explicitly promote flat minima. In contrast, our work focuses on understanding the implicit regularization effects inherent within the standard zeroth-order optimization. In addition to the trace of Hessian used in this work, other notions of sharpness have also been studied in the literature. One such example is the largest eigenvalue of the Hessian matrix, which has been shown to be implicitly penalized by (S)GD with large learning rates [20, 7, 87, 67, 23, 2] and SAM [89, 11]. Li et al. [51] proved that SAM implicitly promotes balanced solutions on scale-invariant problems. A

sharpness measure based on the largest gradient norm in a local neighborhood was proposed in [101]. Recently, Ahn et al. [3] provided a formal definition of flat minima and studied the convergence complexity of finding them. A local concept of flat minima was used, defining them as local minima that are also stationary points of the trace of Hessian evaluated at limit points under gradient flow. Two gradient-based algorithms were proposed with convergence guarantees to flat local minima under the assumptions that the loss function is four times continuously differentiable, satisfies the local PL condition, and has a twice Lipschitz limit map under gradient flow. In this work, we adopt a global notion of flat minima and assume convexity of the function to show that zeroth-order optimization with the two-point estimator converges to flat global minima.

## 2 Warm-up: Sharpness as Implicit Regularization

Throughout this paper, zeroth-order optimization refers specifically to the method described in Algorithm 1. As the two-point estimator in Eq. (1) is an unbiased gradient estimator for the smoothed function $f_\lambda(x) = \mathbb{E}_{u \sim \mathcal{N}(0, \mathrm{I}_d)}[f(x + \lambda u)]$ [74], zeroth-order optimization directly minimizes $f_\lambda(x)$. Let $f(x)$ be twice continuously differentiable. By Taylor's theorem, we have that

$$f(x + \lambda u) = f(x) + \lambda u^\top \nabla f(x) + \frac{\lambda^2}{2} u^\top \nabla^2 f(x) u + o(\lambda^2).$$

Taking expectation w.r.t. $u \sim \mathcal{N}(0, \mathrm{I}_d)$, we obtain that

$$
\begin{aligned}
f_\lambda(x) &= f(x) + \frac{\lambda^2}{2} \mathbb{E}_u \left[ \mathrm{Tr} \left( uu^\top \nabla^2 f(x) \right) \right] + o(\lambda^2) \\
&= f(x) + \frac{\lambda^2}{2} \mathrm{Tr} \left( \nabla^2 f(x) \right) + o(\lambda^2).
\end{aligned}
\tag{2}
$$

The results suggest that the smoothed function introduces trace of Hessian as an additional regularization term. In the literature for sharpness-aware minimization, the trace of Hessian is often used to measure the sharpness of the solution [89, 88, 3]. Recall that $\mathbb{E}[g_\lambda(x, u)] = \nabla f_\lambda(x)$, and thus zeroth-order optimization implicitly minimizes sharpness:

$$\mathbb{E}_{u_t}[x_{t+1} - x_t] = -\eta \nabla f(x_t) - \frac{\eta \lambda^2}{2} \nabla \mathrm{Tr} \left( \nabla^2 f(x_t) \right) + o(\lambda^2). \tag{3}$$

This holds for any twice continuously differentiable function without further assumptions. It can be readily deduced from Eq. (3) that when $x_t$ is close to optimal, i.e., with small gradient, the iterates move in expectation towards a direction that reduces trace of Hessian. Before formally establishing that Eq. (3) leads to flat minima, we first illustrate this intuition through a concrete example.

**Example 2.1.** Consider the function $h(x) = (y^\top z - 1)^2 / 2$, where $x = (y^\top, z^\top)^\top \in \mathbb{R}^{2d}$ for $y, z \in \mathbb{R}^d$. The optimal value is achieved when $y^\top z = 1$, and the trace of Hessian is $\|y\|^2 + \|z\|^2$. Among all optimal solutions, the smallest trace of Hessian is achieved when $y = z$ and $\|y\| = 1$.

Figure 1 plots the values of the loss function and the trace of Hessian when applying gradient descent and zeroth-order optimization on Example 2.1. Gradient descent and zeroth-order optimization with $\lambda \to 0$ leave the trace of Hessian almost unchanged throughout. The smoothing parameter $\lambda$ controls the trade-offs between regularization on the trace of Hessian and the optimization error induced by this additional bias term. Despite the large oscillation in loss values from zeroth-order methods due to random search directions, the trajectory of the trace of Hessian decreases noticeably smoother. Example 2.1 belongs to a class of scale-invariant problems studied in Li et al. [51]. It was proved that SAM promotes balanced solutions on these problems where $B_t := (\|y_t\|^2 - \|z_t\|^2)/2$ converges to 0 in the limit, while $B_t$ remains $B_0$ for gradient descent [51]. Note that only flat minima are perfectly balanced with $\|y\| = \|z\|$ among all optimal solutions of Example 2.1. We show in the following that zeroth-order optimization favors balanced solutions as well. A proof is provided in Appendix A.

**Proposition 2.2.** *When applying zeroth-order optimization (Algorithm 1) on Example 2.1, the limiting flow with $\eta \to 0$ satisfies that $d\mathbb{E}[B_t]/dt = -2\lambda^2 \mathbb{E}[B_t]$. In other words, $\mathbb{E}[B_t] \to 0$ when $t \to \infty$.*

The smoothing parameter $\lambda$ plays a critical role in driving $\mathbb{E}[B_t]$ towards zero and determines the rate. When $\lambda = 0$, the regularization effect disappears, and zeroth-order optimization behaves like gradient descent to maintain $\mathbb{E}[B_t]$ as a constant. These theoretical findings align with Eq. (3) and Figure 1. However, the qualitative discussion of implicit regularization in this section does not offer insights on the complexity to reach flat minima, which will be the subject of the next section.

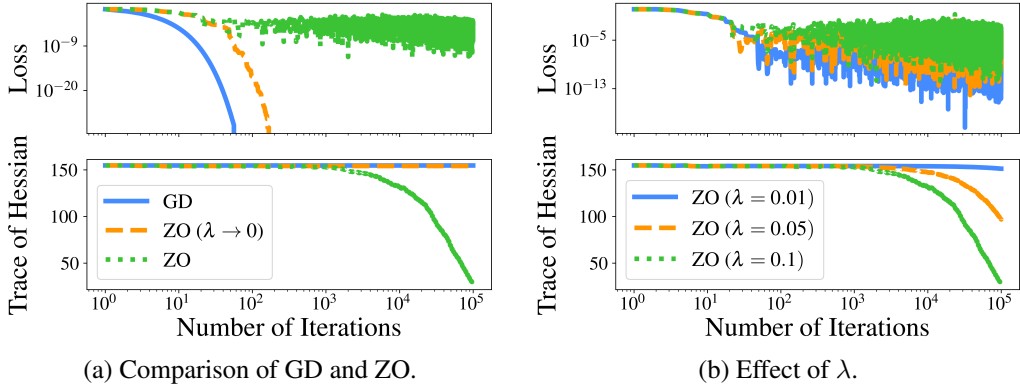

Figure 1: Loss and trace of Hessian on Example 2.1 with $d = 100$. (a) Comparisons among gradient descent (GD), zeroth-order (ZO) optimization (Algorithm 1) with $\lambda = 0.1$, and ZO with $\lambda \to 0$ (directional derivatives). (b) Comparisons on different $\lambda$ used in ZO. GD and ZO with $\lambda \to 0$ fail to decrease the trace of Hessian. Larger $\lambda$ in ZO leads to larger errors in the loss but brings more regularization effect on minimizing the trace of Hessian.

## 3   Complexity for Finding Flat Minima

To formally study the convergence complexity of zeroth-order optimization with the two-point estimator towards flat minima, we first define the notion of flat minima that we are interested in.

**Definition 3.1** (Flat Minima). Suppose that $f(x)$ is twice differentiable with a well-defined Hessian matrix. Let $\mathcal{X}^* := \arg\min_{x \in \mathbb{R}^d} f(x)$ denote the set of minimizers of $f(x)$. We say $x^*$ is a flat minimum of $f(x)$ if $x^* \in \arg\min_{x \in \mathcal{X}^*} \mathrm{Tr}(\nabla^2 f(x))$. That is, $x^*$ is a minimizer of $f(x)$ that achieves the smallest trace of Hessian among the set of minimizers.

The above definition of flat minima implicitly encodes a constrained optimization problem

$$\min_{x \in \mathbb{R}^d} \mathrm{Tr}(\nabla^2 f(x)), \quad \text{s.t.} \quad f(x) - \min_{x \in \mathbb{R}^d} f(x) \leq 0.$$

This functional constraint is equivalent to requiring $x \in \mathcal{X}^*$, since it always holds that $f(x) \geq \min_{x \in \mathbb{R}^d} f(x)$. In the literature on functional constrained optimization problems $\min_{x \in \mathcal{C}} \psi_0(x)$ with the constrained set $\mathcal{C} := \{x \in \mathbb{R}^d \,|\, \psi_1(x) \leq 0\}$, a common objective is to find $\epsilon$-approximate solutions $\hat{x}$ satisfying $\psi_0(\hat{x}) - \min_{x \in \mathcal{C}} \psi_0(x) \leq \epsilon$ and $\psi_1(\hat{x}) \leq \epsilon$ [71, 13, 100]. Motivated from this connection to constrained optimization, we define approximate flat minima in the following.

**Definition 3.2** (Approximate Flat Minima). Suppose that $f(x)$ is twice differentiable with a well-defined Hessian matrix. Let $\mathcal{X}^* := \arg\min_{x \in \mathbb{R}^d} f(x)$ denote the set of minimizers of $f(x)$. For $\epsilon_1, \epsilon_2 > 0$, we say $\hat{x}$ is an $(\epsilon_1, \epsilon_2)$-approximate flat minimum of $f(x)$ if

$$f(\hat{x}) - \min_{x \in \mathbb{R}^d} f(x) \leq \epsilon_1, \quad \mathrm{Tr}(\nabla^2 f(\hat{x})) - \min_{x \in \mathcal{X}^*} \mathrm{Tr}(\nabla^2 f(x)) \leq \epsilon_2.$$

Ahn et al. [3] defined flat minima as local minima that are also stationary points of the trace of Hessian evaluated at limit points under gradient flow. Since the trace of Hessian is highly nonconvex, such a local definition does not necessarily correspond to minima with the lowest trace of Hessian. In this work, we adopt a global notion of flat minima and prove that zeroth-order optimization converges to them. The following assumptions are required to guarantee convergence to approximate flat minima.

**Assumption 3.3** (Smoothness). We assume the function $f(x)$ is three times continuously differentiable and satisfies that $(a)$ $f(x)$ is $L_1$-smooth; $(b)$ $f(x)$ is second-order smooth with $L_2 > 0$, which implies that all third-order partial derivatives are bounded: $|\partial^3 f(x)/\partial x_i \partial x_j \partial x_k| \leq L_2, \forall i, j, k \in [d]$ and $\forall x \in \mathbb{R}^d$; and, $(c)$ $f(x)$ is third-order smooth with $L_3 > 0$, which implies that $\forall x, y \in \mathbb{R}^d$,

$$\left| f(y) - f(x) - \nabla f(x)^\top (y - x) - \frac{1}{2}(y - x)^\top \nabla^2 f(x)(y - x) - \frac{1}{6}\nabla^3 f(x)[y - x]^3 \right| \leq \frac{L_3}{24}\|x - y\|^4,$$

where $\nabla^3 f(x)[y - x]^3 = \sum_{i,j,k \in [d]}(\partial^3 f(x)/\partial x_i \partial x_j \partial x_k)(y_i - x_i)(y_j - x_j)(y_k - x_k)$.

Since the characterization of flat minima already involves second-order information of $f(x)$, it is natural to require assumptions on higher-order information to establish convergence guarantees. Higher-order smoothness assumptions have been widely used to establish fast convergence rates [8, 72], guarantee convergence to second-order stationary points [30, 79], analyze implicit regularization [7, 89], and study the complexity of finding flat minima [3]. We emphasize that these assumptions on higher-order information are used solely for convergence analysis; Algorithm 1 requires only zeroth-order information. In order to prove global convergence, we also need the following convexity assumption. Although $f(x)$ is convex, $\text{Tr}(\nabla^2 f(x))$ is in general nonconvex. Seeking flat minima with lowest $\text{Tr}(\nabla^2 f(x))$ in the set of minimizers $\mathcal{X}^*$ is therefore a challenging task.

**Assumption 3.4** (Convexity). The function $f(x)$ is convex on $\mathbb{R}^d$, and thus $\text{Tr}(\nabla^2 f(x)) \geq 0$.

## 3.1 Convergence Analysis

We start with a brief recap of standard convergence analysis for zeroth-order optimization. We then explain the major theoretical contribution in this paper that leads to convergence towards flat minima. By the zeroth-order updates in Algorithm 1, we have that $\forall x \in \mathbb{R}^d$,

$$
\begin{aligned}
\mathbb{E}\|x_{t+1} - x\|^2 &= \mathbb{E}\|x_t - x\|^2 - 2\eta \, \mathbb{E}[g_\lambda(x_t, u_t)]^\top (x_t - x) + \eta^2 \, \mathbb{E}\|g_\lambda(x_t, u_t)\|^2 \\
&\leq \mathbb{E}\|x_t - x\|^2 - 2\eta(f_\lambda(x_t) - f_\lambda(x)) + \eta^2 \, \mathbb{E}\|g_\lambda(x_t, u_t)\|^2.
\end{aligned}
\tag{4}
$$

Here, we use $\mathbb{E}[g_\lambda(x_t, u_t)] = \nabla f_\lambda(x_t)$ and the property that $f_\lambda(x)$ is convex when $f(x)$ is convex [74]. Standard analysis considers optimizing $f(x)$. When $f(x)$ is smooth, the second term $f_\lambda(x_t) - f_\lambda(x) = f(x_t) - f(x) + \mathcal{O}(\lambda^2)$, and the third term can be bounded as

$$
\mathbb{E}\|g_\lambda(x_t, u_t)\|^2 \leq 2(d+4)\|\nabla f(x)\|^2 + \mathcal{O}(\lambda^2).
$$

Since $f(x)$ is $L_1$-smooth, we also have that $\|\nabla f(x)\|^2 \leq 2L_1(f(x) - \min_{x \in \mathbb{R}^d} f(x))$. By selecting $x$ as one minimizer from the set $\mathcal{X}^*$ and setting $\eta = \mathcal{O}(1/d)$, we can rearrange Eq. (4) to obtain

$$
\mathbb{E}\left[f(x_t) - \min_{x \in \mathbb{R}^d} f(x)\right] \leq \mathcal{O}(d)(\mathbb{E}\|x_t - x\|^2 - \mathbb{E}\|x_{t+1} - x\|^2) + \mathcal{O}(\lambda^2).
$$

Summing up from $t = 0$ to $t = T - 1$ and averaging by $T$ give $\mathbb{E}[f(\bar{x}_T) - \min_{x \in \mathbb{R}^d} f(x)] \leq \mathcal{O}(d/T)$ with a small enough $\lambda$, where $\bar{x}_T$ is the average of iterates [74].

Going beyond the classical analysis and targeting at flat minima, we take inspiration from Eq. (2) and instead focus on the regularized loss

$$
F(x) := f(x) + \frac{\lambda^2}{2}\text{Tr}\left(\nabla^2 f(x)\right).
\tag{5}
$$

In this way, we do not treat $\text{Tr}(\nabla^2 f(x))$ as a bias to be controlled but instead view the term as an objective to be optimized. Indeed, the $\mathcal{O}(\lambda^2)$ error term in the standard analysis mostly comes from bounding $\lambda^2 \text{Tr}(\nabla^2 f(x))$ from above by $\lambda^2 L_1 d$ when $f(x)$ is $L_1$-smooth. To proceed, we need to control the difference between $F(x)$ and $f_\lambda(x)$, as well as to bound $\mathbb{E}\|g_\lambda(x_t, u_t)\|^2$ by the term $\|\nabla F(x)\|^2$ to establish convergence on $F(x)$.

**Lemma 3.5.** *Let Assumption 3.3 be satisfied. For $F(x)$ defined in Eq. (5), it holds that*

$$
|f_\lambda(x) - F(x)| \leq \frac{L_3}{24}\lambda^4(d+4)^2, \qquad \forall x \in \mathbb{R}^d.
$$

*The second moments of the two-point estimator $g_\lambda(x, u)$ defined in Eq. (1) can be bounded as*

$$
\mathbb{E}\|g_\lambda(x, u)\|^2 \leq 2(d+6)\|\nabla F(x)\|^2 + \frac{L_2^2}{3}\lambda^4(d+6)^4 + \frac{L_3^2}{288}\lambda^6(d+10)^5.
$$

Note that the result above is a non-trivial extension of classical analysis. Particularly for $\mathbb{E}\|g_\lambda(x, u)\|^2$, Isserlis' theorem [41, 91] is required to compute an 8th-order moment $\mathbb{E}[\prod_{j=1}^8 u_{i_j}]$ where each $i_j \in [d]$ and $u_{i_j}$ is a standard Gaussian (see Lemma B.1), while standard theory only needs to compute a 4th-order moment. By a combinatorial computation, there are in total 105 terms to be considered for 8th-order moments, but only three terms for 4th-order moments. We also need smoothness of $F(x)$ for relating $\|\nabla F(x)\|^2$ to $F(x) - \min_{x \in \mathbb{R}^d} F(x)$.

**Lemma 3.6.** *Let Assumption 3.3 be satisfied. Then $F(x)$ is $(2L_1)$-smooth if $\lambda^2 \leq \sqrt{2}L_1/(d^{3/2}L_3)$.*

Proofs of Lemmas 3.5 and 3.6 can be found in Appendix B.2. Following the one-step analysis in Eq. (4) and using the above two lemmas, the theorem below establishes convergence guarantees of Algorithm 1 to minima of $F(x)$. A proof is provided in Appendix B.3.

**Theorem 1.** *Under Assumptions 3.3 and 3.4, Algorithm 1 with stepsize $\eta = 1/(8(d+6)L_1)$ and smoothing parameter $\lambda^2 \leq \sqrt{2}L_1/(d^{3/2}L_3)$ satisfies that*

$$\mathbb{E}\left[F(x_\tau) - \min_{x \in \mathbb{R}^d} F(x)\right] \leq \frac{8(d+6)L_1\|x_0 - x_F^*\|^2}{T}$$
$$+ \frac{L_3}{6}\lambda^4(d+4)^2 + \frac{L_2^2}{24L_1}\lambda^4(d+6)^3 + \frac{L_3^2}{1152L_1}\lambda^6(d+10)^4,$$

*where $x_0 \in \mathbb{R}^d$ is the initialization, $x_F^* \in \arg\min_{x \in \mathbb{R}^d} F(x)$, and the expectation is taken w.r.t. the randomness in all search directions and the selection of $x_\tau$.*

Following the standard gradient-based method to minimize $F(x)$ via $\nabla F(x)$, third-order derivatives of $f(x)$ are required due to the computation of $\nabla \text{Tr}(\nabla^2 f(x))$. Here, we prove that Algorithm 1 that uses only zeroth-order information of $f(x)$ converges to minimizers of $F(x)$. Although $F(x)$ is nonconvex, the difference between $F(x)$ and a convex function, $f_\lambda(x)$, can be made small, and thus convergence to global minima is still guaranteed. The corollary below transfers convergence guarantee towards minima of $F(x)$ to convergence guarantees towards flat minima of $f(x)$. The proof of it can be found in Appendix B.3.

**Corollary 2** (Iteration Complexity for Finding Flat Minima). *Let Assumptions 3.3 and 3.4 be satisfied. For $\epsilon > 0$, Algorithm 1 with stepsize $\eta = 1/(8(d+6)L_1)$ satisfies that*

$$\mathbb{E}\left[f(x_\tau) - \min_{x \in \mathbb{R}^d} f(x)\right] \leq \frac{3L_1^2\epsilon}{2L_2^2(d+6)^2}\left(1 + \frac{\epsilon}{L_1(d+6)}\right),$$
$$\mathbb{E}\left[\text{Tr}(\nabla^2 f(x_\tau)) - \min_{x \in \mathcal{X}^*} \text{Tr}(\nabla^2 f(x))\right] \leq \epsilon,$$

*when setting the smoothing parameter $\lambda$ and the number of iterations $T$ such that*

$$\lambda^2 = \min\left\{\frac{\sqrt{2}L_1}{d^{3/2}L_3}, \frac{12\sqrt{L_1\epsilon}}{L_3(d+10)^2}, \frac{3\epsilon}{4L_3(d+4)^2}, \frac{3L_1\epsilon}{L_2^2(d+6)^3}\right\},$$
$$T \geq \frac{64(d+6)L_1\|x_0 - x_F^*\|^2}{\epsilon}\max\left\{\frac{d^{3/2}L_3}{\sqrt{2}L_1}, \frac{L_3(d+10)^2}{12\sqrt{L_1\epsilon}}, \frac{4L_3(d+4)^2}{3\epsilon}, \frac{L_2^2(d+6)^3}{3L_1\epsilon}\right\}.$$

*Recall $\mathcal{X}^* = \arg\min_{x \in \mathbb{R}^d} f(x)$. According to Definition 3.2, this means that $(\mathcal{O}(\epsilon/d^2), \epsilon)$ approximate flat minima can be guaranteed with $T = \mathcal{O}(d^4/\epsilon^2)$ and $\lambda = \mathcal{O}(\epsilon^{1/2}/d^{3/2})$.*

*Remark* 3.1. Our convergence guarantees require that $f(x)$ is three times continuously differentiable and satisfies first-, second-, and third-order smoothness assumptions. In Appendix C, we explain how the first- and third-order smoothness assumptions can be relaxed, at the cost of slower convergence rates; see Table 1 for a summary.

*Remark* 3.2. We discuss here how the parameters $\lambda$ and $T$ are selected and compare with the standard analysis [74]. Let $\mathbb{E}[F(x_\tau) - \min_{x \in \mathbb{R}^d} F(x)] \leq \text{Err}(\lambda, T)$, where $\text{Err}(\lambda, T) = \mathcal{O}(d/T) + \mathcal{O}(\lambda^4 d^3)$ is given by Theorem 1. In the proof of Corollary 2, we show that $\mathbb{E}[f(x_\tau) - \min_{x \in \mathbb{R}^d} f(x)] \leq \text{Err}(\lambda, T) + \mathcal{O}(\lambda^2 d)$ and $\mathbb{E}[\text{Tr}(\nabla^2 f(x_\tau)) - \min_{x \in \mathcal{X}^*} \text{Tr}(\nabla^2 f(x))] \leq 2\text{Err}(\lambda, T)/\lambda^2$. The guarantees on $\mathbb{E}[f(x_\tau) - \min_{x \in \mathbb{R}^d} f(x)] \leq \mathcal{O}(d/T) + \mathcal{O}(\lambda^2 d)$ share the same dependence on the leading error terms as the standard analysis [74], but we additionally ensure $\mathbb{E}[\text{Tr}(\nabla^2 f(x_\tau)) - \min_{x \in \mathcal{X}^*} \text{Tr}(\nabla^2 f(x))] \leq \mathcal{O}(d/(\lambda^2 T)) + \mathcal{O}(\lambda^2 d^3)$. We choose $\lambda^2 = \mathcal{O}(\epsilon/d^3)$ and $T = \mathcal{O}(d/(\lambda^2\epsilon))$ such that $\mathbb{E}[\text{Tr}(\nabla^2 f(x_\tau)) - \min_{x \in \mathcal{X}^*} \text{Tr}(\nabla^2 f(x))] \leq \epsilon$, which results in $\mathbb{E}[f(x_\tau) - \min_{x \in \mathbb{R}^d} f(x)] \leq \mathcal{O}(\lambda^2(\epsilon + d)) = \mathcal{O}(\epsilon/d^2)$. Standard analysis [74] set $T = \mathcal{O}(d/\epsilon)$ and $\lambda^2 = \mathcal{O}(\epsilon/d)$ such that only $\mathbb{E}[f(x_\tau) - \min_{x \in \mathbb{R}^d} f(x)] \leq \epsilon$ holds.

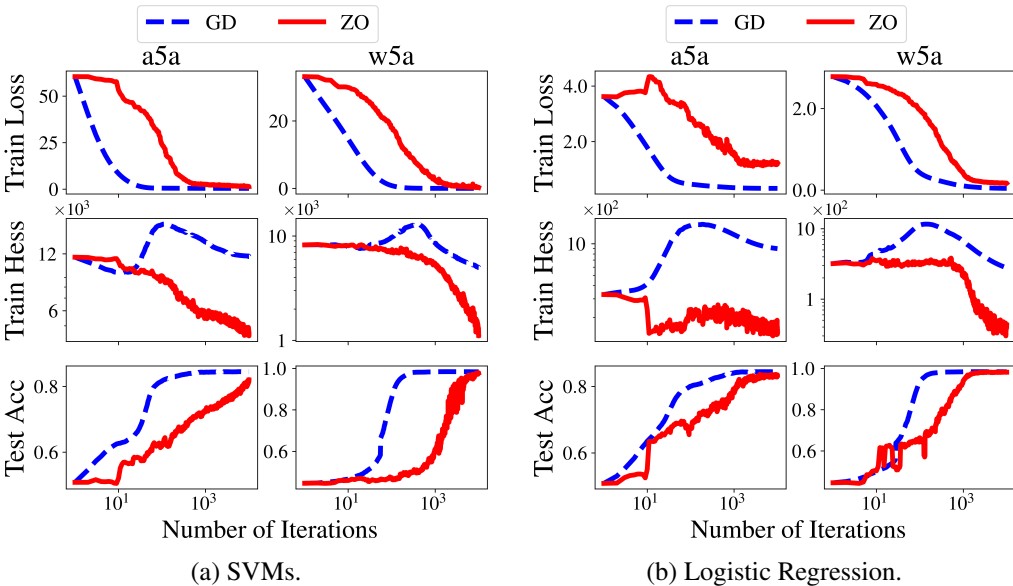

(a) SVMs.                          (b) Logistic Regression.

Figure 2: Training loss, trace of Hessian on the training data (Train Hess in the plot), and test accuracy on (a) SVMs and (b) Logistic regression. Zeroth-order (ZO) optimization is slower than gradient descent (GD) for minimizing the loss and maximizing the accuracy, but there is a clear trend of decreasing trace of Hessian for ZO.

*Remark* 3.3. We focus on the convex setting in this work, but the analysis extends to cases where the objective is locally convex in a neighborhood. When initialized in this region, zeroth-order optimization identifies local minima with the smallest trace of Hessian among all local minimizers in the neighborhood. When $f(x)$ is generally nonconvex, the current definition of flat minima is not theoretically tractable without additional assumptions. We leave for future work a detailed study on the definition of computationally feasible flat minima and the assumptions required to understand the complexity of finding them in the nonconvex setting [2].

*Remark* 3.4. Our proof framework can be extended to understand the complexity of first-order methods towards flat minima. For example, a gradient-based method, $x_{t+1} \leftarrow x_t - \eta \nabla f(x_t + \lambda u_t)$ with $u_t \sim \mathcal{N}(0, \mathrm{I}_d)$, that uses a gradient evaluated at the perturbed point as the descent direction also minimizes the smoothed loss $f_\lambda(x)$ in the expectation. By upper bounding $\mathbb{E}\|\nabla f(x + \lambda u)\|^2$ with the term $\|\nabla F(x)\|^2$, convergence guarantees on $F(x)$ and thus to flat minima can be established. The same framework provides new insights on how SAM can be analyzed as well. Our primary focus is on zeroth-order methods, and extensions to first-order methods are deferred to future.

## 4   Experiments

We provide empirical results on binary classification tasks with convex losses and language model fine-tuning tasks. Our code is available at `https://github.com/Liang137/FlatZero`.

**Binary Classification with SVMs and Logistic Regression.** We start empirical evaluations using support vector machines (SVMs) and logistic regression. Given a training dataset $\{a_i, b_i\}_{i=1}^N$ with feature vectors $a_i \in \mathbb{R}^d$ and binary labels $b_i$, we consider an overparameterized regime where each $a_i \in \mathbb{R}^d$ is mapped to $\phi(a_i) = W a_i \in \mathbb{R}^D$ via a random matrix $W \in \mathbb{R}^{D \times d}$ with $W_{ij} \sim \mathcal{N}(0, 1)$ and $D > N > d$. We use two standard binary classification benchmarks from the LIBSVM library [14]: a5a ($N = 6,414$, $d = 123$) and w5a ($N = 9,888$, $d = 300$) [76], and set $D = 10,000$ to overparameterize. This ensures a higher-dimensional solution set with increased diversity, enabling a meaningful study of implicit regularization to investigate the solutions selected by the algorithm. We

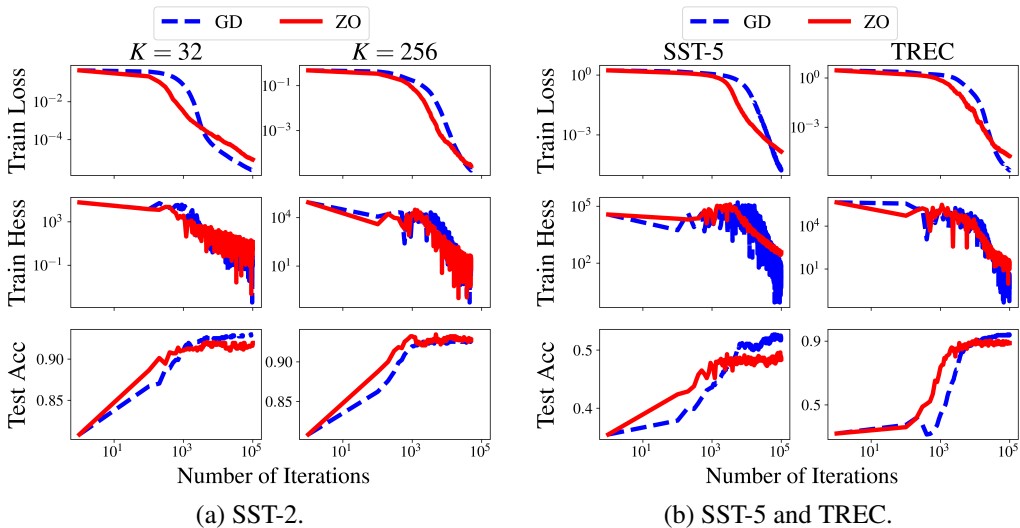

Figure 3: Training loss, trace of Hessian on the training data (Train Hess in the plot), and test accuracy on (a) SST-2 with $K = 32$ and $K = 256$, and (b) SST-5 and TREC with $K = 32$. Both gradient descent (GD) and zeroth-order (ZO) optimization reduce the trace of Hessian during training. In most cases, GD achieves lower training loss, smaller trace of Hessian, and higher test accuracy.

consider SVMs with the squared hinge loss and $b_i \in \{-1, 1\}$, that is,

$$\min_{x \in \mathbb{R}^D} f(x) = \min_{x \in \mathbb{R}^D} \frac{1}{N} \sum_{i=1}^{N} (\max\{0, 1 - b_i \phi(a_i)^\top x\})^2.$$

For $\sigma(z) = 1/(1 + \exp(-z))$ and $b_i \in \{0, 1\}$, logistic regression minimizes the loss

$$\min_{x \in \mathbb{R}^D} f(x) = \min_{x \in \mathbb{R}^D} \frac{1}{N} \sum_{i=1}^{N} \left(-b_i \log \sigma(\phi(a_i)^\top x) - (1 - b_i) \log(1 - \sigma(\phi(a_i)^\top x))\right).$$

These two models can be viewed as linear probing [48] on a two-layer neural network with a frozen first layer and an identity activation. In the SVMs case, the second layer uses an identity activation and is trained with the squared hinge loss, while logistic regression applies a sigmoid activation and minimizes the cross-entropy loss. Both SVMs and logistic regression have convex objective functions, and the trace of Hessian can be efficiently computed. Figure 2 shows the optimization trajectories of gradient descent and zeroth-order optimization. In all cases, gradient descent and zeroth-order optimization achieve comparable training loss and test accuracy; however, zeroth-order optimization consistently reduces the trace of Hessian and converges to flatter solutions. Detailed experimental setups and additional results can be found in Appendix D.2.

**Fine-Tuning Language Models on Text Classification Tasks.** We also evaluate the behaviors of zeroth-order optimization on nonconvex language model fine-tuning tasks. Following Malladi et al. [69], we consider few-shot fine-tuning on RoBERTa-Large (355M parameters) [61] with $K = 32$ and $K = 256$ examples per class on three sentence classification datasets: SST-2 and SST-5 [82] for sentiment classification, and TREC [84] for topic classification. All experiments are tested on a single NVIDIA H100 GPU with 80 GiB memory. To mitigate the effect of mini-batch noise on reducing the trace of Hessian, we use full-batch training for both gradient descent and zeroth-order optimization, which is feasible in the few-shot setting. As the exact computation of the trace of Hessian is intractable due to the large model size, we adopt the widely-used expected sharpness $(2/\delta^2)|\mathbb{E}_{u \sim \mathcal{N}(0, \mathrm{I}_d)}[f(x + \delta u)] - f(x)|$ as an approximation [75, 105, 85], where $f(x)$ denotes the training loss evaluated at model weights $x \in \mathbb{R}^d$. We set $\delta = 10^{-4}$ and estimate the expectation by averaging over 100 samples. The performance of gradient descent and zeroth-order optimization is presented in Figure 3. In this setting, both methods are observed to decrease the trace of Hessian. It is conjectured that the behavior in gradient descent results from the implicit regularization associated

with large learning rates [20, 7]. Meanwhile, the observed decrease in zeroth-order optimization matches our theoretical insights. Detailed setups and additional results are deferred to Appendix D.3.

In classical optimization theory [74, 28], the convergence rates of zeroth-order methods scale with the dimension $d$, limiting their applicability to problems with large $d$ such as language model fine-tuning. Previous work [69, 99] explaining why zeroth-order methods still achieve reasonable performance on language model fine-tuning tasks has relaxed the dependence on $d$ to a term related to the trace of Hessian, $\mathrm{Tr}(\nabla^2 f(x))$. Assuming $\mathrm{Tr}(\nabla^2 f(x)) \ll dL_1$ when $f(x)$ is $L_1$-smooth, zeroth-order optimization achieves dimension-independent rates and remains effective even in high-dimensional settings. Our experimental results show that the trace of Hessian decreases and attains values much smaller than the actual dimension, thereby supporting the assumption made in prior work to explain the empirical success of zeroth-order optimization.

## 5   Conclusion

Motivated by the observation that zeroth-order optimization with the two-point estimator (Algorithm 1) inherently minimizes $\mathrm{Tr}(\nabla^2 f(x))$, we initiate a formal study of this implicit regularization. Specifically, we analyze its convergence to flat minima, defined as the ones with the lowest trace of Hessian among all minimizers. For convex and sufficiently smooth (Assumptions 3.3) functions, we prove that Algorithm 1 guarantees $(\mathcal{O}(\epsilon/d^2), \epsilon)$-approximate flat minima (Definition 3.2) after $T = \mathcal{O}(d^4/\epsilon^2)$ iterations. This is the first work showing that zeroth-order optimization converges to flat minima. Experiments on binary classification tasks using SVMs and logistic regression, as well as language model fine-tuning tasks on RoBERTa support our theoretical findings.

Theoretical and empirical performance of zeroth-order methods is often limited by the high variance in gradient estimation. A promising direction is to combine zeroth-order and first-order methods to leverage the strengths of both [69, 102, 54]. We only examine zeroth-order optimization using the standard two-point estimator. Exploring whether the convergence complexity can be further improved with possible modifications and additional algorithmic designs remains an interesting line of work. The current theoretical results require convexity and higher-order smoothness assumptions of the function. Investigation into nonconvex functions with relaxed assumptions is left for future work.

## Acknowledgments and Disclosure of Funding

We thank all anonymous reviewers for their valuable suggestions. L.Z. gratefully acknowledges funding by the Max Planck ETH Center for Learning Systems. B.L. is supported by Swiss National Science Foundation Project Funding No. 200021-207343. This work does not relate to the current position of K.T. at Amazon. S.O. is supported in part by the National Science Foundation under grant no. 2112471, 2229876, and 2505865 supported in part by funds provided by the National Science Foundation, by the Department of Homeland Security, and by IBM. Any opinions, findings, and conclusions or recommendations expressed in this material are those of the author(s) and do not necessarily reflect the views of the National Science Foundation or its federal agency and industry partners. M.M. is supported by the German Research Foundation. N.H. is supported by ETH research grant funded through ETH Zurich Foundations and Swiss National Science Foundation Project Funding No. 200021-207343.

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

# A Missing Proofs from Section 2

*Proof of Proposition 2.2.* Algorithm 1 applied to Example 2.1 gives

$$\frac{x_{t+1} - x_t}{\eta} = -\frac{h(x_t + \lambda u_t) - h(x_t - \lambda u_t)}{2\lambda} u_t,$$

where $u_t \sim \mathcal{N}(0, \mathrm{I}_{2d})$ is the search direction. Let $u_t = (v_t^\top, w_t^\top)^\top$. The dynamics can be written as

$$\frac{y_{t+1} - y_t}{\eta} = -\frac{\left((y_t + \lambda v_t)^\top (z_t + \lambda w_t) - 1\right)^2 - \left((y_t - \lambda v_t)^\top (z_t - \lambda w_t) - 1\right)^2}{4\lambda} v_t,$$

$$\frac{z_{t+1} - z_t}{\eta} = -\frac{\left((y_t + \lambda v_t)^\top (z_t + \lambda w_t) - 1\right)^2 - \left((y_t - \lambda v_t)^\top (z_t - \lambda w_t) - 1\right)^2}{4\lambda} w_t.$$

Let $\Delta_t := ((y_t + \lambda v_t)^\top (z_t + \lambda w_t) - 1)^2 - ((y_t - \lambda v_t)^\top (z_t - \lambda w_t) - 1)^2$. As it holds that

$$\Delta_t = 4\lambda \left(y_t^\top z_t - 1 + \lambda^2 v_t^\top w_t\right)(y_t^\top w_t + z_t^\top v_t),$$

the dynamics can be simplified as

$$\begin{aligned}
\mathbb{E}\left[\frac{y_{t+1} - y_t}{\eta} \,\middle|\, x_t\right] &= -\mathbb{E}[(y_t^\top z_t - 1 + \lambda^2 v_t^\top w_t)(y_t^\top w_t + z_t^\top v_t)v_t \mid x_t] \\
&= -(y_t^\top z_t - 1)\mathbb{E}[v_t v_t^\top]z_t - \lambda^2 \mathbb{E}[v_t v_t^\top w_t w_t^\top]y_t \\
&= -(y_t^\top z_t - 1)z_t - \lambda^2 y_t,
\end{aligned}$$

and the same reasoning applies to

$$\begin{aligned}
\mathbb{E}\left[\frac{z_{t+1} - z_t}{\eta} \,\middle|\, x_t\right] &= -\mathbb{E}[(y_t^\top z_t - 1 + \lambda^2 v_t^\top w_t)(y_t^\top w_t + z_t^\top v_t)w_t \mid x_t] \\
&= -(y_t^\top z_t - 1)y_t - \lambda^2 z_t.
\end{aligned}$$

Here, we use that $w_t, v_t$ are independent, and that $\mathbb{E}[w_t w_t^\top] = \mathbb{E}[v_t v_t^\top] = \mathrm{I}_d$. For $B_t = (\|y_t\|^2 - \|z_t\|^2)/2$, we then have that

$$\frac{B_{t+1} - B_t}{\eta} = y_t^\top \left(\frac{y_{t+1} - y_t}{\eta}\right) - z_t^\top \left(\frac{z_{t+1} - z_t}{\eta}\right) + \frac{\eta}{2}\left(\frac{y_{t+1} - y_t}{\eta}\right)^2 - \frac{\eta}{2}\left(\frac{z_{t+1} - z_t}{\eta}\right)^2.$$

Since it holds that

$$\begin{aligned}
y_t^\top \mathbb{E}\left[\frac{y_{t+1} - y_t}{\eta} \,\middle|\, x_t\right] - z_t^\top \mathbb{E}\left[\frac{z_{t+1} - z_t}{\eta} \,\middle|\, x_t\right] &= \lambda^2(\|z_t\|^2 - \|y_t\|^2) \\
&= -2\lambda^2 B_t,
\end{aligned}$$

we conclude that

$$\mathbb{E}\left[\frac{B_{t+1} - B_t}{\eta} \,\middle|\, x_t\right] = -2\lambda^2 B_t + \mathcal{O}(\eta).$$

Therefore, by taking the limit $\eta \to 0$, we have that $d\,\mathbb{E}[B_t]/dt = -2\lambda^2\,\mathbb{E}[B_t]$. $\qquad\square$

# B Missing Proofs from Section 3

We use Isserlis' theorem [41, 91] frequently in this section and restate it below for clarity.

*Fact B.1 (Isserlis' theorem).* Let $u \sim \mathcal{N}(0, \mathrm{I}_d)$ be a standard Gaussian random vector in $\mathbb{R}^d$. The $i$-th coordinate of $u$ is denoted by $u_i$. For $i_j \in [d]$ with $j \in [N]$, it holds that

$$\mathbb{E}\left[\prod_{j=1}^N u_{i_j}\right] = \sum_{p \in P_N^2} \prod_{(k,l) \in p} \delta_{i_k i_l},$$

where $P_N^2$ is the set of all distinct ways of partitioning $[N]$ into pairs, and $\delta_{i_k i_l}$ is the Kronecker delta, equal to 1 if $i_k = i_l$, and 0 otherwise.

For example, when $N = 4$, there are in total $\binom{4}{2}\binom{2}{2}/2! = 3$ distinct ways, and we have that

$$P_4^2 = \{\{(1,2),(3,4)\}, \{(1,3),(2,4)\}, \{(1,4),(2,3)\}\}.$$

Applying Isserlis' theorem gives that

$$\mathbb{E}\left[\prod_{j=1}^{4} u_{i_j}\right] = \delta_{i_1 i_2}\delta_{i_3 i_4} + \delta_{i_1 i_3}\delta_{i_2 i_4} + \delta_{i_1 i_4}\delta_{i_2 i_3}.$$

In general, there is no such partitions and the expectation is 0 when $N$ is odd. For the even case, the number of distinct partitions is

$$\frac{\binom{N}{2}\binom{N-2}{2}\binom{N-4}{2}\cdots\binom{4}{2}\binom{2}{2}}{(N/2)!} = (N-1)(N-3)\cdots 3 \cdot 1,$$

which results in $(N-1)!!$ terms in the summation.

## B.1  Supporting Lemmas

Let $f(x)$ be three times continuously differentiable and $u \sim \mathcal{N}(0, \mathrm{I}_d)$ be a standard Gaussian random vector in $\mathbb{R}^d$. To simplify notation in this section, we denote

$$G_0(x, u) = \left(u^\top \nabla f(x)\right) u,$$
$$H_0(x, u) = \left(u^\top \nabla(u^\top \nabla^2 f(x)u)\right) u.$$

Since $\mathbb{E}[uu^\top] = \mathrm{I}_d$, we have that

$$\begin{aligned}\mathbb{E}[u^\top \nabla f(x)u] &= \mathbb{E}[uu^\top]\nabla f(x) \\ &= \nabla f(x).\end{aligned}$$

This shows that the directional derivative $G_0(x, u)$ when taking $\lambda \to 0$ in the two-point estimator (see Eq. (1)) is an unbiased estimator of $\nabla f(x)$. For $H_0(x, u)$, it holds that

$$\begin{aligned}H_0(x, u) &= u \sum_{k=1}^{d} u_k \frac{\partial}{\partial x_k}(u^\top \nabla^2 f(x)u) \\ &= u \sum_{k=1}^{d} u_k \frac{\partial}{\partial x_k}\left(\sum_{i=1}^{d}\sum_{j=1}^{d}\frac{\partial^2 f(x)}{\partial x_i \partial x_j}u_i u_j\right) \\ &= u \sum_{i=1}^{d}\sum_{j=1}^{d}\sum_{k=1}^{d}\frac{\partial^3 f(x)}{\partial x_i \partial x_j \partial x_k}u_i u_j u_k.\end{aligned}$$

Let $h_l$ be the $l$-th coordinate of $H_0(x, u)$ for $l \in [d]$. Applying Isserlis' theorem, we have that

$$\begin{aligned}\mathbb{E}[h_l] &= \sum_{i=1}^{d}\sum_{j=1}^{d}\sum_{k=1}^{d}\frac{\partial^3 f(x)}{\partial x_i \partial x_j \partial x_k}\mathbb{E}[u_i u_j u_k u_l] \\ &= \sum_{i=1}^{d}\sum_{j=1}^{d}\sum_{k=1}^{d}\frac{\partial^3 f(x)}{\partial x_i \partial x_j \partial x_k}(\delta_{ij}\delta_{kl} + \delta_{ik}\delta_{jl} + \delta_{il}\delta_{jk}) \\ &= 3\frac{\partial}{\partial x_l}\left(\sum_{i=1}^{d}\frac{\partial^2 f(x)}{\partial x_i^2}\right).\end{aligned}$$

This means that $\mathbb{E}[H_0(x, u)] = 3\nabla(\mathrm{Tr}(\nabla^2 f(x)))$, and that $G_0(x, u) + (\lambda^2/6)H_0(x, u)$ is an unbiased gradient estimator for $F(x)$ defined in Eq. (5).

**Lemma B.1.** *Let $f(x)$ be three times continuously differentiable and $u \sim \mathcal{N}(0, \mathrm{I}_d)$ be a standard Gaussian random vector in $\mathbb{R}^d$ independent of $x \in \mathbb{R}^d$. It holds that*

*(a)* $\mathbb{E}\|G_0(x, u)\|^2 = (d+2)\|\nabla f(x)\|^2,$

*(b)* $\mathbb{E}[G_0(x, u)^\top H_0(x, u)] = 3(d+4)\,\nabla f(x)^\top \nabla \mathrm{Tr}(\nabla^2 f(x)),$

*(c)* $\mathbb{E}\|H_0(x, u)\|^2 = 9(d+6)\|\nabla \mathrm{Tr}(\nabla^2 f(x))\|^2 + 6(d+6) \sum_{i=1}^{d} \sum_{j=1}^{d} \sum_{k=1}^{d} \left( \dfrac{\partial^3 f(x)}{\partial x_i \partial x_j \partial x_k} \right)^2.$

*This enables the computation of an upper bound on $\mathbb{E}\|G_0(x, u) + (\lambda^2/6)\,H_0(x, u)\|^2$.*

*Proof.* For $(a)$, we have that

$$
\begin{aligned}
\mathbb{E}\big\|u^\top \nabla f(x) u\big\|^2 &= \mathbb{E}\left[ \left( \sum_{i=1}^{d} u_i \frac{\partial f(x)}{\partial x_i} \right)^2 \left( \sum_{i=1}^{d} u_i^2 \right) \right] \\
&= \sum_{i=1}^{d} \sum_{j=1}^{d} \sum_{k=1}^{d} \frac{\partial f(x)}{\partial x_i} \frac{\partial f(x)}{\partial x_j} \mathbb{E}[u_i u_j u_k^2] \\
&= \sum_{i=1}^{d} \sum_{j=1}^{d} \sum_{k=1}^{d} \frac{\partial f(x)}{\partial x_i} \frac{\partial f(x)}{\partial x_j} (\delta_{ij} + 2\delta_{ik}\delta_{jk}) \\
&= (d+2) \sum_{i=1}^{d} \left( \frac{\partial f(x)}{\partial x_i} \right)^2 \\
&= (d+2)\|\nabla f(x)\|^2.
\end{aligned}
$$

Similarly for $(b)$, we have that

$$
\begin{aligned}
\mathbb{E}[G_0(x, u)^\top H_0(x, u)] &= \mathbb{E}\left[ (u^\top \nabla f(x))(u^\top \nabla(u^\top \nabla^2 f(x) u))\|u\|^2 \right] \\
&= \mathbb{E}\left[ \left( \sum_{i=1}^{d} u_i \frac{\partial f(x)}{\partial x_i} \right) \left( \sum_{i=1}^{d} \sum_{j=1}^{d} \sum_{k=1}^{d} \frac{\partial^3 f(x)}{\partial x_i \partial x_j \partial x_k} u_i u_j u_k \right) \left( \sum_{i=1}^{d} u_i^2 \right) \right] \\
&= \sum_{i=1}^{d} \sum_{j=1}^{d} \sum_{k=1}^{d} \sum_{m=1}^{d} \sum_{n=1}^{d} \frac{\partial^3 f(x)}{\partial x_i \partial x_j \partial x_k} \frac{\partial f(x)}{\partial x_m} \mathbb{E}[u_i u_j u_k u_m u_n^2].
\end{aligned}
$$

By Isserlis' theorem (see Fact B.1), when $N = 6$ to compute $\mathbb{E}[u_i u_j u_k u_m u_n^2]$, there are in total $5 \times 3 = 15$ terms in the summation, and the problem reduces to how we partition the set $\{i, j, k, m, n, n\}$ into three pairs. Let us consider the following two cases.

(1) We pair $n$ with $n$ and partition $\{i, j, k, m\}$ into two pairs. This gives 3 terms in the summation.

$$
\delta_{ij}\delta_{km} + \delta_{ik}\delta_{jm} + \delta_{im}\delta_{jk} := A_1.
$$

Here, we use $\delta_{nn} = 1$ and let $A_1$ denote the summation of terms considered in case (1), and then

$$
\sum_{i=1}^{d} \sum_{j=1}^{d} \sum_{k=1}^{d} \sum_{m=1}^{d} \sum_{n=1}^{d} \frac{\partial^3 f(x)}{\partial x_i \partial x_j \partial x_k} \frac{\partial f(x)}{\partial x_m} A_1 = 3d \sum_{i=1}^{d} \sum_{k=1}^{d} \frac{\partial^3 f(x)}{\partial x_i^2 \partial x_k} \frac{\partial f(x)}{\partial x_k}.
$$

(2) We select two ordered indices in $\{i, j, k, m\}$, pair the first one with the first $n$, and pair the second one with the second $n$. The remaining two indices form the third pair. This gives $4 \times 3 = 12$ terms in the summation. To further simplify notation, we divide case (2) into two subgroups, each with 6 terms, and denote the summation of considered terms by $A_{2-1}$, $A_{2-2}$, respectively.

In $A_{2-1}$, we consider the case where both selected indices come from $\{i, j, k\}$ involved in third-order derivatives. It contains 6 terms and thus

$$
\sum_{i=1}^{d} \sum_{j=1}^{d} \sum_{k=1}^{d} \sum_{m=1}^{d} \sum_{n=1}^{d} \frac{\partial^3 f(x)}{\partial x_i \partial x_j \partial x_k} \frac{\partial f(x)}{\partial x_m} A_{2-1} = 6 \sum_{i=1}^{d} \sum_{k=1}^{d} \frac{\partial^3 f(x)}{\partial x_i^2 \partial x_k} \frac{\partial f(x)}{\partial x_k}.
$$

In $A_{2-2}$, we consider the case where one index in the selected two indices is $m$. This also gives 6 terms, and we have that

$$\sum_{i=1}^{d}\sum_{j=1}^{d}\sum_{k=1}^{d}\sum_{m=1}^{d}\sum_{n=1}^{d}\frac{\partial^3 f(x)}{\partial x_i \partial x_j \partial x_k}\frac{\partial f(x)}{\partial x_m}\, A_{2-2} = 6\sum_{i=1}^{d}\sum_{k=1}^{d}\frac{\partial^3 f(x)}{\partial x_i^2 \partial x_k}\frac{\partial f(x)}{\partial x_k}.$$

Considering the above cases $(1)$ and $(2)$, we conclude the proof of $(b)$.

$$\mathbb{E}[G_0(x,u)^\top H_0(x,u)] = \sum_{i=1}^{d}\sum_{j=1}^{d}\sum_{k=1}^{d}\sum_{m=1}^{d}\sum_{n=1}^{d}\frac{\partial^3 f(x)}{\partial x_i \partial x_j \partial x_k}\frac{\partial f(x)}{\partial x_m}\mathbb{E}[u_i u_j u_k u_m u_n^2]$$

$$= \sum_{i=1}^{d}\sum_{j=1}^{d}\sum_{k=1}^{d}\sum_{m=1}^{d}\sum_{n=1}^{d}\frac{\partial^3 f(x)}{\partial x_i \partial x_j \partial x_k}\frac{\partial f(x)}{\partial x_m}(A_1 + A_{2-1} + A_{2-2})$$

$$= 3(d+4)\sum_{i=1}^{d}\sum_{k=1}^{d}\frac{\partial^3 f(x)}{\partial x_i^2 \partial x_k}\frac{\partial f(x)}{\partial x_k}$$

$$= 3(d+4)\,\nabla f(x)^\top \nabla \mathrm{Tr}(\nabla^2 f(x)).$$

Finally for $(c)$, we have that

$$\mathbb{E}\|H_0(x,u)\|^2 = \mathbb{E}\left[(u^\top \nabla(u^\top \nabla^2 f(x)u))^2 \|u\|^2\right]$$

$$= \mathbb{E}\left[\left(\sum_{i=1}^{d}\sum_{j=1}^{d}\sum_{k=1}^{d}\frac{\partial^3 f(x)}{\partial x_i \partial x_j \partial x_k}u_i u_j u_k\right)^2\left(\sum_{i=1}^{d}u_i^2\right)\right]$$

$$= \sum_{i_1,j_1,k_1=1}^{d}\sum_{i_2,j_2,k_2=1}^{d}\sum_{l=1}^{d}\frac{\partial^3 f(x)}{\partial x_{i_1}\partial x_{j_1}\partial x_{k_1}}\frac{\partial^3 f(x)}{\partial x_{i_2}\partial x_{j_2}\partial x_{k_2}}\mathbb{E}[u_{i_1}u_{j_1}u_{k_1}u_{i_2}u_{j_2}u_{k_2}u_l^2],$$

where $\sum_{i_1,j_1,k_1=1}^{d}$ means $\sum_{i_1=1}^{d}\sum_{j_1=1}^{d}\sum_{k_1=1}^{d}$. By Isserlis' theorem (see Fact B.1), we need to consider all distinct ways of partitioning the set $\{i_1,j_1,k_1,i_2,j_2,k_2,l,l\}$ into four pairs, which has $7 \times 5 \times 3 = 105$ terms in the summation. We consider the following two cases.

$(1)$ We pair $l$ with $l$ and partition $\{i_1,j_1,k_1,i_2,j_2,k_2\}$ into three pairs. This gives $5 \times 3 = 15$ terms in the summation. We further divide case $(1)$ into two subgroups and denote the summation of considered terms by $A_{1-1}$, $A_{1-2}$, respectively.

$A_{1-1}$ has $3 \times 3 = 9$ terms, where the first pair is formed with one index from $\{i_1,j_1,k_1\}$ and the other index from $\{i_2,j_2,k_2\}$, the remaining indices in $\{i_1,j_1,k_1\}$ yield the second pair, and the remaining indices in $\{i_2,j_2,k_2\}$ give the third pair. By symmetry of the summation, we have that

$$\sum_{i_1,j_1,k_1=1}^{d}\sum_{i_2,j_2,k_2=1}^{d}\sum_{l=1}^{d}\frac{\partial^3 f(x)}{\partial x_{i_1}\partial x_{j_1}\partial x_{k_1}}\frac{\partial^3 f(x)}{\partial x_{i_2}\partial x_{j_2}\partial x_{k_2}}\, A_{1-1} = 9d\sum_{i=1}^{d}\sum_{j=1}^{d}\sum_{k=1}^{d}\frac{\partial^3 f(x)}{\partial x_i^2 \partial x_k}\frac{\partial^3 f(x)}{\partial x_j^2 \partial x_k}.$$

The rest 6 terms of cases $(1)$ are collected in $A_{1-2}$, where all three pairs consist of one index from $\{i_1,j_1,k_1\}$ and the other index from $\{i_2,j_2,k_2\}$. Therefore, we have that

$$\sum_{i_1,j_1,k_1=1}^{d}\sum_{i_2,j_2,k_2=1}^{d}\sum_{l=1}^{d}\frac{\partial^3 f(x)}{\partial x_{i_1}\partial x_{j_1}\partial x_{k_1}}\frac{\partial^3 f(x)}{\partial x_{i_2}\partial x_{j_2}\partial x_{k_2}}\, A_{1-2} = 6d\sum_{i=1}^{d}\sum_{j=1}^{d}\sum_{k=1}^{d}\left(\frac{\partial^3 f(x)}{\partial x_i \partial x_j \partial x_k}\right)^2.$$

$(2)$ We select two ordered indices in $\{i_1,j_1,k_1,i_2,j_2,k_2\}$, pair the first one with the first $l$, and pair the second one with the second $l$. The remaining four indices form the last two pairs. This gives $(6 \times 5) \times 3 = 90$ terms in the summation. We further divide case $(2)$ into two subgroups and denote the summation of considered terms by $A_{2-1}$, $A_{2-2}$, respectively.

In $A_{2-1}$, the selected two indices come from the same $\{i_1,j_1,k_1\}$ or $\{i_2,j_2,k_2\}$. There are in total $3 \times 2 \times 2 \times 3 = 36$ terms involved. By symmetry of the summation, we have that

$$\sum_{i_1,j_1,k_1=1}^{d}\sum_{i_2,j_2,k_2=1}^{d}\sum_{l=1}^{d}\frac{\partial^3 f(x)}{\partial x_{i_1}\partial x_{j_1}\partial x_{k_1}}\frac{\partial^3 f(x)}{\partial x_{i_2}\partial x_{j_2}\partial x_{k_2}}\, A_{2-1} = 36\sum_{i=1}^{d}\sum_{j=1}^{d}\sum_{k=1}^{d}\frac{\partial^3 f(x)}{\partial x_i^2 \partial x_k}\frac{\partial^3 f(x)}{\partial x_j^2 \partial x_k}.$$

In $A_{2-2}$, the selected two indices consist of one from $\{i_1, j_1, k_1\}$ and the other from $\{i_2, j_2, k_2\}$ to pair with $l$. To simplify computation, $A_{2-2}$ is further split into $A_{2-2-1}$ and $A_{2-2-2}$, according to how the remaining two pairs are constructed.

For case $A_{2-2-1}$, after having the two pairs with $l$, the remaining indices in $\{i_1, j_1, k_1\}$ yield the third pair, and the remaining indices in $\{i_2, j_2, k_2\}$ give the fourth pair. One example of such partition is $\{(i_1, l), (i_2, l), (j_1, k_1), (j_2, k_2)\}$. There are in total $3 \times 3 \times 2 = 18$ terms in the summation, and

$$
\sum_{i_1,j_1,k_1=1}^{d} \sum_{i_2,j_2,k_2=1}^{d} \sum_{l=1}^{d} \frac{\partial^3 f(x)}{\partial x_{i_1} \partial x_{j_1} \partial x_{k_1}} \frac{\partial^3 f(x)}{\partial x_{i_2} \partial x_{j_2} \partial x_{k_2}} A_{2-2-1} = 18 \sum_{i=1}^{d}\sum_{j=1}^{d}\sum_{k=1}^{d} \frac{\partial^3 f(x)}{\partial x_i^2 \partial x_k} \frac{\partial^3 f(x)}{\partial x_j^2 \partial x_k}.
$$

For case $A_{2-2-2}$, after having the two pairs with $l$, the remaining two pairs are composed of one index from $\{i_1, j_1, k_1\}$ and the other index from $\{i_2, j_2, k_2\}$. There are $3 \times 3 \times 2 \times 2 = 36$ terms, and thus we have that

$$
\sum_{i_1,j_1,k_1=1}^{d} \sum_{i_2,j_2,k_2=1}^{d} \sum_{l=1}^{d} \frac{\partial^3 f(x)}{\partial x_{i_1} \partial x_{j_1} \partial x_{k_1}} \frac{\partial^3 f(x)}{\partial x_{i_2} \partial x_{j_2} \partial x_{k_2}} A_{2-2-2} = 36 \sum_{i=1}^{d}\sum_{j=1}^{d}\sum_{k=1}^{d} \left( \frac{\partial^3 f(x)}{\partial x_i \partial x_j \partial x_k} \right)^2.
$$

A sanity check $36+18+36 = 90$ makes sure that all terms are included in case (2), and $15+90 = 105$ ensures all terms are considered to compute $\mathbb{E}[u_{i_1} u_{j_1} u_{k_1} u_{i_2} u_{j_2} u_{k_2} u_l^2]$. Therefore, considering all cases above, we have that

$$
\mathbb{E}\|H_0(x,u)\|^2 = \sum_{i_1,j_1,k_1=1}^{d} \sum_{i_2,j_2,k_2=1}^{d} \sum_{l=1}^{d} \frac{\partial^3 f(x)}{\partial x_{i_1} \partial x_{j_1} \partial x_{k_1}} \frac{\partial^3 f(x)}{\partial x_{i_2} \partial x_{j_2} \partial x_{k_2}} \mathbb{E}[u_{i_1} u_{j_1} u_{k_1} u_{i_2} u_{j_2} u_{k_2} u_l^2]
$$

$$
= 9(d+6) \sum_{i=1}^{d}\sum_{j=1}^{d}\sum_{k=1}^{d} \frac{\partial^3 f(x)}{\partial x_i^2 \partial x_k} \frac{\partial^3 f(x)}{\partial x_j^2 \partial x_k} + 6(d+6) \sum_{i=1}^{d}\sum_{j=1}^{d}\sum_{k=1}^{d} \left( \frac{\partial^3 f(x)}{\partial x_i \partial x_j \partial x_k} \right)^2
$$

$$
= 9(d+6)\|\nabla \mathrm{Tr}(\nabla^2 f(x))\|^2 + 6(d+6) \sum_{i=1}^{d}\sum_{j=1}^{d}\sum_{k=1}^{d} \left( \frac{\partial^3 f(x)}{\partial x_i \partial x_j \partial x_k} \right)^2.
$$

This proves $(c)$ and concludes the proof of Lemma B.1. $\qquad\square$

## B.2  Proofs of Lemmas 3.5 and 3.6

*Proof of Lemma 3.5.* Recall that $f_\lambda(x) = \mathbb{E}[f(x + \lambda u)]$ and $F(x) = f(x) + (\lambda^2/2)\,\mathrm{Tr}(\nabla^2 f(x))$. The first part follows from the third-order smoothness of $f(x)$ in Assumption 3.3. To simplify notation, we let

$$
\Phi_x(x + \lambda u) = f(x) + \lambda \nabla f(x)^\top u + \frac{\lambda^2}{2} u^\top \nabla^2 f(x) u + \frac{\lambda^3}{6} \nabla^3 f(x)[u]^3, \tag{6}
$$

where $\nabla^3 f(x)[u]^3 = \sum_{i,j,k \in [d]} (\partial^3 f(x)/\partial x_i \partial x_j \partial x_k) u_i u_j u_k$ and $u \sim \mathcal{N}(0, I_d)$ is a standard Gaussian random vector. Taking expectation w.r.t. $u$, we have that

$$
\mathbb{E}_u[\Phi_x(x + \lambda u)] = f(x) + \frac{\lambda^2}{2} \mathbb{E}[u^\top \nabla^2 f(x) u]
$$

$$
= F(x).
$$

As a result, we obtain

$$
|f_\lambda(x) - F(x)| = \left| \mathbb{E}[f(x + \lambda u)] - \mathbb{E}[\Phi_x(x + \lambda u)] \right|
$$

$$
\leq \mathbb{E}\left| f(x + \lambda u) - \Phi_x(x + \lambda u) \right|
$$

$$
\leq \frac{L_3}{24} \lambda^4 \mathbb{E}\|u\|^4,
$$

where we use Assumption 3.3 $(c)$. The proof is complete applying Lemma 1 in Nesterov and Spokoiny [74] such that $\mathbb{E}\|u\|^\gamma \leq (d + \gamma)^{\gamma/2}$ if $\gamma \geq 2$.

We now show the second part using Lemma B.1. Similarly to Eq. (6), we define

$$\Phi_x(x - \lambda u) = f(x) - \lambda \nabla f(x)^\top u + \frac{\lambda^2}{2} u^\top \nabla^2 f(x) u - \frac{\lambda^3}{6} \nabla^3 f(x)[u]^3.$$

By Assumption 3.3 $(c)$, we have that

$$|f(x + \lambda u) - f(x - \lambda u)| \leq |f(x + \lambda u) - \Phi_x(x + \lambda u)| + |\Phi_x(x - \lambda u) - f(x - \lambda u)|$$
$$+ |\Phi_x(x + \lambda u) - \Phi_x(x - \lambda u)|$$
$$\leq \left| 2\lambda \nabla f(x)^\top u + \frac{\lambda^3}{3} \nabla^3 f(x)[u]^3 \right| + \frac{L_3}{12} \lambda^4 \|u\|^4.$$

For the two-point estimator in Eq. (1), it holds that

$$\|g_\lambda(x, u)\|^2 = \frac{(f(x + \lambda u) - f(x - \lambda u))^2}{4\lambda^2} \|u\|^2$$
$$\leq 2 \left( \nabla f(x)^\top u + \frac{\lambda^2}{6} \nabla^3 f(x)[u]^3 \right)^2 \|u\|^2 + \frac{L_3^2}{288} \lambda^6 \|u\|^{10}.$$

Recall $G_0(x, u) = uu^\top \nabla f(x)$ and $H_0(x, u) = uu^\top \nabla(u^\top \nabla^2 f(x) u) = u \nabla^3 f(x)[u]^3$. Therefore, we have that

$$\mathbb{E}\|g_\lambda(x, u)\|^2$$
$$\leq 2\mathbb{E}\|G_0(x, u)\|^2 + \frac{2\lambda^2}{3} \mathbb{E}[G_0(x, u)^\top H_0(x, u)] + \frac{\lambda^4}{18} \mathbb{E}\|H_0(x, u)\|^2 + \frac{L_3^2 \lambda^6}{288} \mathbb{E}\|u\|^{10}$$
$$= 2(d + 2)\|\nabla f(x)\|^2 + 2(d + 4)\lambda^2 \nabla f(x)^\top \nabla \mathrm{Tr}(\nabla^2 f(x)) + \frac{(d + 6)\lambda^4}{2} \|\nabla \mathrm{Tr}(\nabla^2 f(x))\|^2$$
$$+ \frac{(d + 6)\lambda^4}{3} \sum_{i=1}^{d} \sum_{j=1}^{d} \sum_{k=1}^{d} \left( \frac{\partial^3 f(x)}{\partial x_i \partial x_j \partial x_k} \right)^2 + \frac{L_3^2 \lambda^6}{288} \mathbb{E}\|u\|^{10}$$
$$\leq 2(d + 6)\|\nabla F(x)\|^2 + \frac{L_2^2}{3} \lambda^4 (d + 6)^4 + \frac{L_3^2}{288} \lambda^6 (d + 10)^5.$$

Here, we plug in Lemma B.1, apply Assumption 3.3 $(b)$, and use Lemma 1 in [74].

For completeness, we also show here that second-order smoothness implies the boundedness of third-order derivatives. Recall the second-order smoothness in Assumption 3.3 $(b)$.

$$\|\nabla^2 f(y) - \nabla^2 f(z)\| \leq L_2 \|y - z\|, \quad \forall y, z \in \mathbb{R}^d,$$

where the matrix norm is defined as

$$\|\nabla^2 f(y) - \nabla^2 f(z)\| := \max_{w \in \mathbb{R}^d, \|w\| \leq 1} \left| w^\top \nabla^2 f(y) w - w^\top \nabla^2 f(z) w \right|.$$

For simplicity of the notation, we let $\forall i, j \in [d]$,

$$D_{ij}(y, z) := \frac{\partial^2 f(y)}{\partial x_i \partial x_j} - \frac{\partial^2 f(z)}{\partial x_i \partial x_j}.$$

Let $e_i \in \mathbb{R}^d$ denote the unit vector whose $i$-th coordinate is 1 and all other coordinates are 0. Since $\|e_i\| = 1$, we have that $\forall i \in [d]$,

$$|D_{ii}(y, z)| = \left| e_i^\top \nabla^2 f(y) e_i - e_i^\top \nabla^2 f(z) e_i \right|$$
$$\leq \|\nabla^2 f(y) - \nabla^2 f(z)\|$$
$$\leq L_2 \|y - z\|.$$

We define $w^+ := (e_i + e_j)/\sqrt{2}$ and $w^- := (e_i - e_j)/\sqrt{2}$ for any $i, j \in [d]$ and $i \neq j$. It holds that

$$(w^+)^\top (\nabla^2 f(y) - \nabla^2 f(z))(w^+) = \frac{1}{2}(D_{ii}(y, z) + 2D_{ij}(y, z) + D_{jj}(y, z)),$$
$$(w^-)^\top (\nabla^2 f(y) - \nabla^2 f(z))(w^-) = \frac{1}{2}(D_{ii}(y, z) - 2D_{ij}(y, z) + D_{jj}(y, z)).$$

Since $\|w^+\| = \|w^-\| = 1$, we have that

$$
\begin{aligned}
|D_{ij}(y,z)| &= \frac{1}{2}\left|(w^+)^\top(\nabla^2 f(y) - \nabla^2 f(z))(w^+) - (w^-)^\top(\nabla^2 f(y) - \nabla^2 f(z))(w^-)\right| \\
&\leq \|\nabla^2 f(y) - \nabla^2 f(z)\| \\
&\leq L_2\|y-z\|.
\end{aligned}
$$

This means that the function $\partial^2 f(x)/\partial x_i \partial x_j$ is $L_2$-Lipschitz for all $i,j \in [d]$ and implies that its gradient norm is upper bounded by $L_2$. As a result, we have that $\forall i,j,k \in [d]$,

$$
\begin{aligned}
\left(\frac{\partial^3 f(x)}{\partial x_i \partial x_j \partial x_k}\right)^2 &\leq \left\|\nabla\left(\frac{\partial^2 f(x)}{\partial x_i \partial x_j}\right)\right\|^2 \\
&\leq L_2^2.
\end{aligned}
$$

This concludes the proof that $|\partial^3 f(x)/\partial x_i \partial x_j \partial x_k| \leq L_2, \forall i,j,k \in [d]$. $\qquad\square$

*Proof of Lemma 3.6.* Recall the third-order smoothness in Assumption 3.3 (*c*).

$$
\|\nabla^3 f(y) - \nabla^3 f(z)\| \leq L_3\|y-z\|, \quad \forall y,z \in \mathbb{R}^d,
$$

where the tensor norm is defined as

$$
\|\nabla^3 f(y) - \nabla^3 f(z)\| := \max_{w \in \mathbb{R}^d, \|w\| \leq 1}\left|\nabla^3 f(y)[w]^3 - \nabla^3 f(z)[w]^3\right|,
$$

and $\nabla^3 f(y)[w]^3 = \sum_{i,j,k \in [d]}(\partial^3 f(y)/\partial x_i \partial x_j \partial x_k)w_i w_j w_k$. More details can be found in Eq. (1.1)–(1.5) of Nesterov [72] and Appendix 1 of Nesterov and Nemirovskii [73]. For simplicity of notation, we let $\forall i,j,k \in [d]$,

$$
D_{ijk}(y,z) := \frac{\partial^3 f(y)}{\partial x_i \partial x_j \partial x_k} - \frac{\partial^3 f(z)}{\partial x_i \partial x_j \partial x_k}.
$$

Let $e_i \in \mathbb{R}^d$ denote the unit vector whose $i$-th coordinate is 1 and all other coordinates are 0. Since $\|e_i\| = 1$, we have that $\forall i \in [d]$,

$$
\begin{aligned}
|D_{iii}(y,z)| &= \left|\nabla^3 f(y)[e_i]^3 - \nabla^3 f(z)[e_i]^3\right| \\
&\leq \|\nabla^3 f(y) - \nabla^3 f(z)\| \\
&\leq L_3\|y-z\|.
\end{aligned}
$$

We define $w^+ := (e_i + e_k)/\sqrt{2}$ and $w^- := (e_i - e_k)/\sqrt{2}$ for any $i,k \in [d]$ and $i \neq k$. It holds that

$$
\nabla^3 f(y)[w^+]^3 - \nabla^3 f(z)[w^+]^3 = \frac{\sqrt{2}}{4}(D_{iii}(y,z) + 3D_{iik}(y,z) + 3D_{kki}(y,z) + D_{kkk}(y,z)),
$$

$$
\nabla^3 f(y)[w^-]^3 - \nabla^3 f(z)[w^-]^3 = \frac{\sqrt{2}}{4}(D_{iii}(y,z) - 3D_{iik}(y,z) + 3D_{kki}(y,z) - D_{kkk}(y,z)).
$$

Since $\|w^+\| = \|w^-\| = 1$, we have that

$$
\begin{aligned}
&|D_{iik}(y,z)| \\
&= \frac{\sqrt{2}}{3}\left|(\nabla^3 f(y)[w^+]^3 - \nabla^3 f(z)[w^+]^3) - (\nabla^3 f(y)[w^-]^3 - \nabla^3 f(z)[w^-]^3) - \frac{\sqrt{2}}{2}D_{kkk}(y,z)\right| \\
&\leq \frac{2\sqrt{2}}{3}\|\nabla^3 f(y) - \nabla^3 f(z)\| + \frac{1}{3}|D_{kkk}(y,z)| \\
&\leq \sqrt{2}L_3\|y-z\|.
\end{aligned}
$$

As a result, we obtain

$$
\begin{aligned}
\|\nabla \mathrm{Tr}(\nabla^2 f(y)) - \nabla \mathrm{Tr}(\nabla^2 f(z))\|^2 &= \sum_{k=1}^{d} \left( \frac{\partial}{\partial x_k} \sum_{i=1}^{d} \frac{\partial^2 f(y)}{\partial x_i^2} - \frac{\partial}{\partial x_k} \sum_{i=1}^{d} \frac{\partial^2 f(z)}{\partial x_i^2} \right)^2 \\
&= \sum_{k=1}^{d} \left( \sum_{i=1}^{d} D_{iik}(y, z) \right)^2 \\
&\leq \sum_{k=1}^{d} \left( \sum_{i=1}^{d} |D_{iik}(y, z)| \right)^2 \\
&\leq 2d^3 L_3^2 \|y - z\|^2.
\end{aligned}
$$

By Assumption 3.3 $(a)$, we have that $\forall y, z \in \mathbb{R}^d$,

$$
\begin{aligned}
\|\nabla F(y) - \nabla F(z)\| &\leq \|\nabla f(y) - \nabla f(z)\| + \frac{\lambda^2}{2} \|\nabla \mathrm{Tr}(\nabla^2 f(y)) - \nabla \mathrm{Tr}(\nabla^2 f(z))\| \\
&\leq \left( L_1 + \frac{\lambda^2 d^{3/2} L_3}{\sqrt{2}} \right) \|y - z\|.
\end{aligned}
$$

This means that $F(x)$ is $(2L_1)$-smooth when $\lambda^2 \leq \sqrt{2} L_1 / (d^{3/2} L_3)$. Therefore, we have that

$$
\begin{aligned}
F\left( x - \frac{1}{2L_1} \nabla F(x) \right) &\leq F(x) - \frac{1}{2L_1} \|\nabla F(x)\|^2 + L_1 \left\| \frac{1}{2L_1} \nabla F(x) \right\|^2 \\
&= F(x) - \frac{1}{4L_1} \|\nabla F(x)\|^2.
\end{aligned}
$$

Rearranging terms, we obtain that $\forall x \in \mathbb{R}^d$,

$$
\begin{aligned}
\|\nabla F(x)\|^2 &\leq 4L_1 \left( F(x) - F\left( x - \frac{1}{2L_1} \nabla F(x) \right) \right) \\
&\leq 4L_1 \left( F(x) - \min_{x \in \mathbb{R}^d} F(x) \right).
\end{aligned}
\tag{7}
$$

This property of smooth functions is used in the proof of Theorem 1 below. $\qquad \square$

### B.3 Proofs of Theorem 1 and Corollary 2

*Proof of Theorem 1.* By Assumption 3.4, $f(x)$ is convex, which suggests that the smoothed function $f_\lambda(x)$ is also convex [74]. Let $x_F^*$ be one minimizer of $F(x) = f(x) + (\lambda^2/2)\mathrm{Tr}(\nabla^2 f(x))$. By the zeroth-order update rule in Algorithm 1, we have that

$$
\begin{aligned}
\mathbb{E}_{u_t} \|x_{t+1} - x_F^*\|^2 &= \|x_t - x_F^*\|^2 - 2\eta \, \mathbb{E}[g_\lambda(x_t, u_t)]^\top (x_t - x_F^*) + \eta^2 \, \mathbb{E}\|g_\lambda(x_t, u_t)\|^2 \\
&= \|x_t - x_F^*\|^2 - 2\eta \nabla f_\lambda(x_t)^\top (x_t - x_F^*) + \eta^2 \, \mathbb{E}\|g_\lambda(x_t, u_t)\|^2 \\
&\leq \|x_t - x_F^*\|^2 - 2\eta(f_\lambda(x_t) - f_\lambda(x_F^*)) + \eta^2 \, \mathbb{E}\|g_\lambda(x_t, u_t)\|^2 \\
&\leq \|x_t - x_F^*\|^2 - 2\eta(F(x_t) - F(x_F^*)) + \eta^2 \, \mathbb{E}\|g_\lambda(x_t, u_t)\|^2 \\
&\quad + 2\eta|F(x_t) - f_\lambda(x_t)| + 2\eta|f_\lambda(x_F^*) - F(x_F^*)| \\
&\leq \|x_t - x_F^*\|^2 - 2\eta(F(x_t) - F(x_F^*)) + \eta^2 \, \mathbb{E}\|g_\lambda(x_t, u_t)\|^2 + \frac{L_3}{6}\eta\lambda^4(d+4)^2,
\end{aligned}
$$

where we apply Lemma 3.5 in the last step. Using the same lemma, we also have that

$$
\begin{aligned}
\mathbb{E}\|g_\lambda(x_t, u_t)\|^2 &\leq 2(d+6)\|\nabla F(x_t)\|^2 + \frac{L_2^2}{3}\lambda^4(d+6)^4 + \frac{L_3^2}{288}\lambda^6(d+10)^5 \\
&\leq 8(d+6)L_1(F(x_t) - F(x_F^*)) + \frac{L_2^2}{3}\lambda^4(d+6)^4 + \frac{L_3^2}{288}\lambda^6(d+10)^5,
\end{aligned}
$$

where we apply Lemma 3.6 with $\lambda^2 \leq \sqrt{2}L_1/(d^{3/2}L_3)$ and use Eq. (7) when $F(x)$ is $(2L_1)$-smooth. As a result, we obtain that

$$\mathbb{E}_{u_t}\|x_{t+1} - x_F^*\|^2 \leq \|x_t - x_F^*\|^2 - 2\eta(1 - 4\eta(d+6)L_1)(F(x_t) - F(x_F^*))$$
$$+ \frac{L_3}{6}\eta\lambda^4(d+4)^2 + \frac{L_2^2}{3}\eta^2\lambda^4(d+6)^4 + \frac{L_3^2}{288}\eta^2\lambda^6(d+10)^5.$$

We choose $\eta = 1/(8(d+6)L_1)$ such that $1 - 4\eta(d+6)L_1 = 1/2$. Rearranging terms, we have

$$\mathbb{E}[F(x_t) - F(x_F^*)]$$
$$\leq \frac{\mathbb{E}\|x_t - x_F^*\|^2 - \mathbb{E}\|x_{t+1} - x_F^*\|^2}{\eta} + \frac{L_3}{6}\lambda^4(d+4)^2 + \frac{L_2^2}{3}\eta\lambda^4(d+6)^4 + \frac{L_3^2}{288}\eta\lambda^6(d+10)^5$$
$$\leq 8(d+6)L_1(\mathbb{E}\|x_t - x_F^*\|^2 - \mathbb{E}\|x_{t+1} - x_F^*\|^2)$$
$$+ \frac{L_3}{6}\lambda^4(d+4)^2 + \frac{L_2^2}{24L_1}\lambda^4(d+6)^3 + \frac{L_3^2}{1152L_1}\lambda^6(d+10)^4.$$

Summing up from $t = 0$ to $t = T - 1$ and dividing both sides by $T$, we have that

$$\mathbb{E}\left[F(x_\tau) - \min_{x \in \mathbb{R}^d} F(x)\right]$$
$$= \frac{1}{T}\sum_{t=0}^{T-1}\mathbb{E}[F(x_t) - F(x_F^*)]$$
$$\leq \frac{8(d+6)L_1\|x_0 - x_F^*\|^2}{T} + \frac{L_3}{6}\lambda^4(d+4)^2 + \frac{L_2^2}{24L_1}\lambda^4(d+6)^3 + \frac{L_3^2}{1152L_1}\lambda^6(d+10)^4.$$

The proof is thus complete. $\qquad\square$

*Proof of Corollary 2.* First, we show the guarantee on the function $f(x)$. Assumption 3.4 implies that $\text{Tr}(\nabla^2 f(x)) \geq 0, \forall x \in \mathbb{R}^d$, and thus $f(x) \leq F(x), \forall x \in \mathbb{R}^d$. Since $f(x)$ is $L_1$-smooth by Assumption 3.3, we also have that

$$\min_{x \in \mathbb{R}^d} F(x) = \min_{x \in \mathbb{R}^d}\left(f(x) + \frac{\lambda^2}{2}\text{Tr}(\nabla^2 f(x))\right)$$
$$\leq \min_{x \in \mathbb{R}^d}\left(f(x) + \frac{\lambda^2}{2}L_1 d\right)$$
$$= \min_{x \in \mathbb{R}^d} f(x) + \frac{\lambda^2}{2}L_1 d.$$

As a result, we obtain that

$$\mathbb{E}\left[f(x_\tau) - \min_{x \in \mathbb{R}^d} f(x)\right] \leq \mathbb{E}\left[F(x_\tau) - \min_{x \in \mathbb{R}^d} F(x)\right] + \frac{L_1}{2}\lambda^2 d$$
$$\leq \frac{8(d+6)L_1\|x_0 - x_F^*\|^2}{T} + \frac{L_1}{2}\lambda^2 d$$
$$+ \frac{L_3}{6}\lambda^4(d+4)^2 + \frac{L_2^2}{24L_1}\lambda^4(d+6)^3 + \frac{L_3^2}{1152L_1}\lambda^6(d+10)^4.$$

Next, we prove the guarantee on the trace of Hessian. Since $\mathcal{X}^* = \arg\min_{x \in \mathbb{R}^d} f(x)$, we have that

$$\min_{x \in \mathbb{R}^d} F(x) = \min_{x \in \mathbb{R}^d}\left(f(x) + \frac{\lambda^2}{2}\text{Tr}(\nabla^2 f(x))\right)$$
$$\leq \min_{x \in \mathcal{X}^*}\left(f(x) + \frac{\lambda^2}{2}\text{Tr}(\nabla^2 f(x))\right)$$
$$= \min_{x \in \mathbb{R}^d} f(x) + \frac{\lambda^2}{2}\min_{x \in \mathcal{X}^*}\text{Tr}(\nabla^2 f(x)).$$

As a result, we obtain that

$$
\mathbb{E}\left[\mathrm{Tr}(\nabla^2 f(x_\tau)) - \min_{x \in \mathcal{X}^*} \mathrm{Tr}(\nabla^2 f(x))\right]
$$

$$
\leq \frac{2}{\lambda^2}\mathbb{E}[F(x_\tau) - f(x_\tau)] - \frac{2}{\lambda^2}\left(\min_{x \in \mathbb{R}^d} F(x) - \min_{x \in \mathbb{R}^d} f(x)\right)
$$

$$
\leq \frac{2}{\lambda^2}\mathbb{E}\left[F(x_\tau) - \min_{x \in \mathbb{R}^d} F(x)\right]
$$

$$
\leq \frac{16(d+6)L_1 \|x_0 - x_F^*\|^2}{\lambda^2 T} + \frac{L_3}{3}\lambda^2 (d+4)^2 + \frac{L_2^2}{12 L_1}\lambda^2 (d+6)^3 + \frac{L_3^2}{576 L_1}\lambda^4 (d+10)^4.
$$

Given $\epsilon > 0$, choosing $\lambda$ and $T$ such that

$$
\lambda^2 = \min\left\{\frac{\sqrt{2}L_1}{d^{3/2}L_3}, \frac{12\sqrt{L_1\epsilon}}{L_3(d+10)^2}, \frac{3\epsilon}{4L_3(d+4)^2}, \frac{3L_1\epsilon}{L_2^2(d+6)^3}\right\},
$$

$$
T \geq \frac{64(d+6)L_1\|x_0 - x_F^*\|^2}{\epsilon} \max\left\{\frac{d^{3/2}L_3}{\sqrt{2}L_1}, \frac{L_3(d+10)^2}{12\sqrt{L_1\epsilon}}, \frac{4L_3(d+4)^2}{3\epsilon}, \frac{L_2^2(d+6)^3}{3L_1\epsilon}\right\},
$$

to make sure that each term in the right-hand side of Eq. (8) is at most $\epsilon/4$, we have that

$$
\mathbb{E}\left[\mathrm{Tr}(\nabla^2 f(x_\tau)) - \min_{x \in \mathcal{X}^*}\mathrm{Tr}(\nabla^2 f(x))\right] \leq \epsilon.
$$

This also ensures that $\mathbb{E}[F(x_\tau) - \min_{x \in \mathbb{R}^d} F(x)] \leq \lambda^2\epsilon/2$, and thus

$$
\mathbb{E}\left[f(x_\tau) - \min_{x \in \mathbb{R}^d} f(x)\right] \leq \mathbb{E}\left[F(x_\tau) - \min_{x \in \mathbb{R}^d} F(x)\right] + \frac{L_1}{2}\lambda^2 d
$$

$$
\leq \frac{\lambda^2}{2}(\epsilon + L_1 d)
$$

$$
\leq \frac{3L_1^2\epsilon}{2L_2^2(d+6)^2}\left(1 + \frac{\epsilon}{L_1(d+6)}\right).
$$

The proof is thus complete. $\qquad\square$

## C   Alternative Proof of the Main Result

In Section 3.1, we provide a convergence analysis by relating $\mathbb{E}\|g_\lambda(x_t, u_t)\|^2$ to $\|\nabla F(x)\|^2$. Here, we show that it is also possible to first bound $\mathbb{E}\|g_\lambda(x_t, u_t)\|^2$ by $\|\nabla f_\lambda(x)\|^2$.

**Lemma C.1.** *Let Assumption 3.3 be satisfied. The second moments of $g_\lambda(x, u)$ can be bounded as*

$$
\mathbb{E}\|g_\lambda(x, u)\|^2 \leq 4(d+2)\|\nabla f_\lambda(x)\|^2 + \frac{L_2^2}{6}\lambda^4(d+8)^5.
$$

*Proof.* The proof mostly follows from Lemma 3 and Theorem 4 in Nesterov and Spokoiny [74] and is similar to that of Lemma 3.5. With a slight abuse of notation, we define

$$
\Phi_x(x + \lambda u) = f(x) + \lambda \nabla f(x)^\top u + \frac{\lambda^2}{2}u^\top \nabla^2 f(x)u,
$$

$$
\Phi_x(x - \lambda u) = f(x) - \lambda \nabla f(x)^\top u + \frac{\lambda^2}{2}u^\top \nabla^2 f(x)u.
$$

Assumption 3.3 (b) implies that

$$
|f(x + \lambda u) - \Phi_x(x + \lambda u)| \leq \frac{L_2}{6}\lambda^3\|u\|^3,
$$

and the same holds for $|f(x - \lambda u) - \Phi_x(x - \lambda u)|$. Therefore, we have that

$$\begin{aligned}
|f(x + \lambda u) - f(x - \lambda u)| &\leq |f(x + \lambda u) - \Phi_x(x + \lambda u)| + |\Phi_x(x - \lambda u) - f(x - \lambda u)| \\
&\quad + |\Phi_x(x + \lambda u) - \Phi_x(x - \lambda u)| \\
&\leq |2\lambda \nabla f(x)^\top u| + \frac{L_2}{3}\lambda^3 \|u\|^3.
\end{aligned}$$

For the two-point estimator $g_\lambda(x, u)$, it holds that

$$\begin{aligned}
\|g_\lambda(x, u)\|^2 &= \frac{(f(x + \lambda u) - f(x - \lambda u))^2}{4\lambda^2}\|u\|^2 \\
&\leq 2(u^\top \nabla f(x))^2 \|u\|^2 + \frac{L_2^2}{18}\lambda^4 \|u\|^8.
\end{aligned}$$

Recall $G_0(x, u) = uu^\top \nabla f(x)$ in Lemma B.1. By Lemma 1 in [74], we obtain that

$$\begin{aligned}
\mathbb{E}\|g_\lambda(x, u)\|^2 &\leq 2\mathbb{E}\|G_0(x, u)\|^2 + \frac{L_2^2}{18}\lambda^4 \mathbb{E}\|u\|^8 \\
&\leq 2(d + 2)\|\nabla f(x)\|^2 + \frac{L_2^2}{18}\lambda^4(d + 8)^4.
\end{aligned}$$

Then we upper bound the difference $\|\nabla f(x) - \nabla f_\lambda(x)\|$ as

$$\begin{aligned}
\|\nabla f(x) - \nabla f_\lambda(x)\| &= \left\|\mathbb{E}\left[u^\top \nabla f(x)u - \frac{f(x + \lambda u) - f(x - \lambda u)}{2\lambda}u\right]\right\| \\
&\leq \frac{1}{2\lambda}\mathbb{E}\|(\Phi_x(x + \lambda u) - \Phi_x(x - \lambda u) - f(x + \lambda u) + f(x - \lambda u))u\| \\
&\leq \frac{L_2}{6}\lambda^2 \mathbb{E}\|u\|^4 \\
&\leq \frac{L_2}{6}\lambda^2(d + 4)^2.
\end{aligned}$$

As a result, this gives that

$$\begin{aligned}
\mathbb{E}\|g_\lambda(x, u)\|^2 &\leq 4(d + 2)\|\nabla f_\lambda(x)\|^2 + 4(d + 2)\|\nabla f(x) - \nabla f_\lambda(x)\|^2 + \frac{L_2^2}{18}\lambda^4(d + 8)^4 \\
&\leq 4(d + 2)\|\nabla f_\lambda(x)\|^2 + \frac{L_2^2}{6}\lambda^4(d + 8)^5.
\end{aligned}$$

The error term $\mathcal{O}(\lambda^4 d^5)$ has worse dependence on $d$ compared to Lemma 3.5. $\qquad\square$

**Theorem 3.** *Under Assumptions 3.3 and 3.4, Algorithm 1 with stepsize $\eta = 1/(8(d + 2)L_1)$ satisfies*

$$\mathbb{E}\left[F(\bar{x}_T) - \min_{x\in\mathbb{R}^d} F(x)\right] \leq \frac{8(d + 2)L_1\|x_0 - x_\lambda^*\|^2}{T} + \frac{L_2^2}{12L_1}\lambda^4(d + 8)^4 + \frac{L_3}{12}\lambda^4(d + 4)^2.$$

*where $x_0 \in \mathbb{R}^d$ is the initialization, $x_\lambda^* \in \arg\min_{x\in\mathbb{R}^d} f_\lambda(x)$, $\bar{x}_T = (1/T)\sum_{t=0}^{T-1} x_t$ is the average of iterates, and the expectation is taken w.r.t. the randomness in all search directions. This implies*

$$\mathbb{E}\left[f(\bar{x}_T) - \min_{x\in\mathbb{R}^d} f(x)\right] \leq \frac{L_1^2\epsilon}{L_2^2(d + 8)^3}\left(1 + \frac{\epsilon}{L_1(d + 8)}\right),$$

$$\mathbb{E}\left[\text{Tr}(\nabla^2 f(\bar{x}_T)) - \min_{x\in\mathcal{X}^*} \text{Tr}(\nabla^2 f(x))\right] \leq \epsilon,$$

*when setting the smoothing parameter $\lambda$ and the number of iterations $T$ such that*

$$\lambda^2 = \min\left\{\frac{2\epsilon}{L_3(d + 4)^2}, \frac{2L_1\epsilon}{L_2^2(d + 8)^4}\right\},$$

$$T \geq \frac{48(d + 2)L_1\|x_0 - x_\lambda^*\|^2}{\epsilon}\max\left\{\frac{L_3(d + 4)^2}{2\epsilon}, \frac{L_2^2(d + 8)^4}{2L_1\epsilon}\right\}.$$

*According to Definition 3.2, this means that $(\mathcal{O}(\epsilon/d^3), \epsilon)$ approximate flat minima can be guaranteed with $T = \mathcal{O}(d^5/\epsilon^2)$ and $\lambda = \mathcal{O}(\epsilon^{1/2}/d^2)$.*

Table 1: Summary of convergence rates to approximate flat minima with different set of assumptions. Convexity is required for all cases and is thus omitted in the table. $L_0$ means $f(x)$ is $L_0$-Lipschitz, and $L_r$ denotes that $f(x)$ is $r$th-order smooth for $r = 1, 2, 3$. Under the $L_1$-smoothness assumption, $(\mathcal{O}(\lambda^2 d), \epsilon)$ approximate flat minima are attained; without this assumption, zeroth-order optimization converges to $(\mathcal{O}(\lambda d^{1/2}), \epsilon)$ approximate flat minima.

| Assumptions | Reference | Smoothing $\lambda$ | Stepsize $\eta$ | Iteration Complexity |
|---|---|---|---|---|
| $L_1, L_2, L_3$ | Corollary 2 | $\mathcal{O}(\epsilon^{1/2}/d^{3/2})$ | $\mathcal{O}(1/d)$ | $\mathcal{O}(d^4/\epsilon^2)$ |
| $L_1, L_2$ | Remark C.2 | $\mathcal{O}(\min\{\epsilon^{1/2}/d^2, \epsilon/d^{3/2}\})$ | $\mathcal{O}(1/d)$ | $\mathcal{O}(\max\{d^5/\epsilon^2, d^4/\epsilon^3\})$ |
| $L_0, L_2, L_3$ | Remark C.3 | $\mathcal{O}(\min\{\epsilon^{1/2}/d, \epsilon^{1/3}/d^{4/3}\})$ | $\mathcal{O}(\lambda/d)$ | $\mathcal{O}(\max\{d^5/\epsilon^2, d^4/\epsilon^{5/2}\})$ |
| $L_0, L_2$ | Remark C.3 | $\mathcal{O}(\epsilon/d^{3/2})$ | $\mathcal{O}(\lambda/d)$ | $\mathcal{O}(d^{11/2}/\epsilon^4)$ |

*Remark* C.1. Compared to Corollary 2, Theorem 3 has similar dependence on $\epsilon$ but worse dependence on $d$ in number of iterations $T$. This comes from the worse dependence $\mathcal{O}(\lambda^4 d^5)$ in Lemma C.1.

*Remark* C.2. In this framework, the only place that uses the third-order smoothness assumption is to upper bound the difference $|F(x) - f_\lambda(x)|$. With second-order smoothness, an alternative result $|F(x) - f_\lambda(x)| \leq (L_2/6)\lambda^3(d+3)^{3/2}$ has worse dependence on $\lambda$; see Theorem 1 in [74]. This leads to $\mathbb{E}[\text{Tr}(\nabla^2 f(\bar{x}_T)) - \min_{x \in \mathcal{X}^*} \text{Tr}(\nabla^2 f(x))] \leq \mathcal{O}(d/(\lambda^2 T) + \lambda^2 d^4 + \lambda d^{3/2})$ and thus $T = \mathcal{O}(\max\{d^5/\epsilon^2, d^4/\epsilon^3\})$. Relaxing the third-order smoothness assumption gives strictly worse convergence rates. The detailed derivations are similar to the proof below and are thus omitted.

*Remark* C.3. We also briefly discuss how the $L_1$-smoothness assumption on $f(x)$ can be relaxed to the function being differentiable and $L_0$-Lipschitz, i.e., $\|\nabla f(x)\| \leq L_0, \forall x \in \mathbb{R}^d$. By Lemma E.3 in [27], when $f(x)$ is convex and $L_0$-Lipschitz, $f_\lambda(x)$ is convex and $(L_0/\lambda)$-smooth. This means all dependence on $L_1$ can be replaced by $L_0/\lambda$ in the analysis below. With both second-order and third-order smoothness assumptions, and setting $\eta = \mathcal{O}(\lambda/d)$, this gives $\mathbb{E}[\text{Tr}(\nabla^2 f(\bar{x}_T)) - \min_{x \in \mathcal{X}^*} \text{Tr}(\nabla^2 f(x))] \leq \mathcal{O}(d/(\lambda^3 T) + \lambda^3 d^4 + \lambda^2 d^2)$ and thus $T = \mathcal{O}(\max\{d^5/\epsilon^2, d^4/\epsilon^{5/2}\})$. With only second-order smoothness assumption, this gives $\mathbb{E}[\text{Tr}(\nabla^2 f(\bar{x}_T)) - \min_{x \in \mathcal{X}^*} \text{Tr}(\nabla^2 f(x))] \leq \mathcal{O}(d/(\lambda^3 T) + \lambda^3 d^4 + \lambda d^{3/2})$ and thus $T = \mathcal{O}(d^{11/2}/\epsilon^4)$. Table 1 summarizes the complexity under different set of assumptions on the function $f(x)$.

*Proof.* When $f(x)$ is $L_1$-smooth and convex, $f_\lambda(x)$ is also $L_1$-smooth and convex [74]. Let $x_\lambda^*$ be one minimizer of $f_\lambda(x)$. The same as the proof of Theorem 1, we obtain that

$$\mathbb{E}_{u_t}\|x_{t+1} - x_\lambda^*\|^2 = \|x_t - x_\lambda^*\|^2 - 2\eta\,\mathbb{E}[g_\lambda(x_t, u_t)]^\top(x_t - x_\lambda^*) + \eta^2\,\mathbb{E}\|g_\lambda(x_t, u_t)\|^2$$

$$= \|x_t - x_\lambda^*\|^2 - 2\eta\nabla f_\lambda(x_t)^\top(x_t - x_\lambda^*) + \eta^2\,\mathbb{E}\|g_\lambda(x_t, u_t)\|^2$$

$$\leq \|x_t - x_\lambda^*\|^2 - 2\eta(f_\lambda(x_t) - f_\lambda(x_\lambda^*)) + \eta^2\,\mathbb{E}\|g_\lambda(x_t, u_t)\|^2.$$

By Lemma C.1, we have that

$$\mathbb{E}\|g_\lambda(x_t, u_t)\|^2 \leq 4(d+2)\|\nabla f_\lambda(x_t)\|^2 + \frac{L_2^2}{6}\lambda^4(d+8)^5$$

$$\leq 8(d+2)L_1(f_\lambda(x_t) - f_\lambda(x_\lambda^*)) + \frac{L_2^2}{6}\lambda^4(d+8)^5.$$

As a result, we obtain that

$$\mathbb{E}_{u_t}\|x_{t+1} - x_\lambda^*\|^2 \leq \|x_t - x_\lambda^*\|^2 - 2\eta(1 - 4\eta(d+2)L_1)(f_\lambda(x_t) - f_\lambda(x_\lambda^*)) + \frac{L_2^2}{6}\eta^2\lambda^4(d+8)^5.$$

We choose $\eta = 1/(8(d+2)L_1)$ such that $1 - 4\eta(d+2)L_1 = 1/2$. Rearranging terms, we have

$$\mathbb{E}[f_\lambda(x_t) - f_\lambda(x_\lambda^*)] \leq \frac{\mathbb{E}\|x_t - x_\lambda^*\|^2 - \mathbb{E}\|x_{t+1} - x_\lambda^*\|^2}{\eta} + \frac{L_2^2}{6}\eta\lambda^4(d+8)^5$$

$$\leq 8(d+2)L_1(\mathbb{E}\|x_t - x_\lambda^*\|^2 - \mathbb{E}\|x_{t+1} - x_\lambda^*\|^2) + \frac{L_2^2}{12L_1}\lambda^4(d+8)^4.$$

Summing up from $t = 0$ to $t = T - 1$ and dividing both sides by $T$, we have that

$$\mathbb{E}\left[f_\lambda(\bar{x}_T) - \min_{x \in \mathbb{R}^d} f_\lambda(x)\right] \leq \frac{1}{T}\sum_{t=0}^{T-1}\mathbb{E}[f_\lambda(x_t) - f_\lambda(x_\lambda^*)]$$
$$\leq \frac{8(d+2)L_1\|x_0 - x_\lambda^*\|^2}{T} + \frac{L_2^2}{12L_1}\lambda^4(d+8)^4,$$

where $\bar{x}_T = (1/T)\sum_{t=0}^{T-1} x_t$ is the average of iterates. By Lemma 3.5, we have that

$$\min_{x \in \mathbb{R}^d} f_\lambda(x) = \min_{x \in \mathbb{R}^d}(F(x) + f_\lambda(x) - F(x))$$
$$\leq \min_{x \in \mathbb{R}^d}(F(x) + |f_\lambda(x) - F(x)|)$$
$$\leq \min_{x \in \mathbb{R}^d} F(x) + \frac{L_3}{24}\lambda^4(d+4)^2.$$

As a result, we obtain that

$$\mathbb{E}\left[F(\bar{x}_T) - \min_{x \in \mathbb{R}^d} F(x)\right] \leq \mathbb{E}\left[f_\lambda(\bar{x}_T) - \min_{x \in \mathbb{R}^d} f_\lambda(x)\right] + |f_\lambda(\bar{x}_T) - F(\bar{x}_T)| + \min_{x \in \mathbb{R}^d} f_\lambda(x) - \min_{x \in \mathbb{R}^d} F(x)$$
$$\leq \frac{8(d+2)L_1\|x_0 - x_\lambda^*\|^2}{T} + \frac{L_2^2}{12L_1}\lambda^4(d+8)^4 + \frac{L_3}{12}\lambda^4(d+4)^2.$$

According to the proof of Corollary 2 (see Eq. (8)), this implies that

$$\mathbb{E}\left[\mathrm{Tr}(\nabla^2 f(\bar{x}_T)) - \min_{x \in \mathcal{X}^*}\mathrm{Tr}(\nabla^2 f(x))\right] \leq \frac{2}{\lambda^2}\mathbb{E}\left[F(\bar{x}_T) - \min_{x \in \mathbb{R}^d} F(x)\right]$$
$$\leq \frac{16(d+2)L_1\|x_0 - x_\lambda^*\|^2}{\lambda^2 T} + \frac{L_2^2}{6L_1}\lambda^2(d+8)^4 + \frac{L_3}{6}\lambda^2(d+4)^2.$$

Given $\epsilon > 0$, choosing $\lambda$ and $T$ such that

$$\lambda^2 = \min\left\{\frac{2\epsilon}{L_3(d+4)^2}, \frac{2L_1\epsilon}{L_2^2(d+8)^4}\right\},$$
$$T \geq \frac{48(d+2)L_1\|x_0 - x_\lambda^*\|^2}{\epsilon}\max\left\{\frac{L_3(d+4)^2}{2\epsilon}, \frac{L_2^2(d+8)^4}{2L_1\epsilon}\right\},$$

we guarantee that $\mathbb{E}[\mathrm{Tr}(\nabla^2 f(\bar{x}_T)) - \min_{x \in \mathcal{X}^*}\mathrm{Tr}(\nabla^2 f(x))] \leq \epsilon$ and that

$$\mathbb{E}\left[f(\bar{x}_T) - \min_{x \in \mathbb{R}^d} f(x)\right] \leq \mathbb{E}\left[F(\bar{x}_T) - \min_{x \in \mathbb{R}^d} F(x)\right] + \frac{L_1}{2}\lambda^2 d$$
$$\leq \frac{\lambda^2}{2}(\epsilon + L_1 d)$$
$$\leq \frac{L_1^2\epsilon}{L_2^2(d+8)^3}\left(1 + \frac{\epsilon}{L_1(d+8)}\right).$$

The proof is thus complete. $\square$

# D  Additional Experiments

## D.1  Test Function

In Figures 1 and 4, we plot the values of the loss function and the trace of Hessian when applying gradient descent and zeroth-order optimization on Example 2.1. We test the example with dimension $d = 100$ and run the algorithms for $T = 100,000$ iterations. Gradient descent uses a stepsize of $0.01$, and zeroth-order optimization uses $0.001$. In Figures 1 (a) and 4 (a), zeroth-order optimization is run with a smoothing parameter $\lambda = 0.1$. The initialization is sampled from the standard Gaussian distribution $\mathcal{N}(0, I_{2d})$. Randomness arises from both the initialization and random search directions in zeroth-order optimization. Figure 1 uses random seed 13, and Figure 4 reports results for additional seeds 17 and 73. The results are consistent across all 3 seeds. Each run is executed on a CPU and takes approximately 1 second.

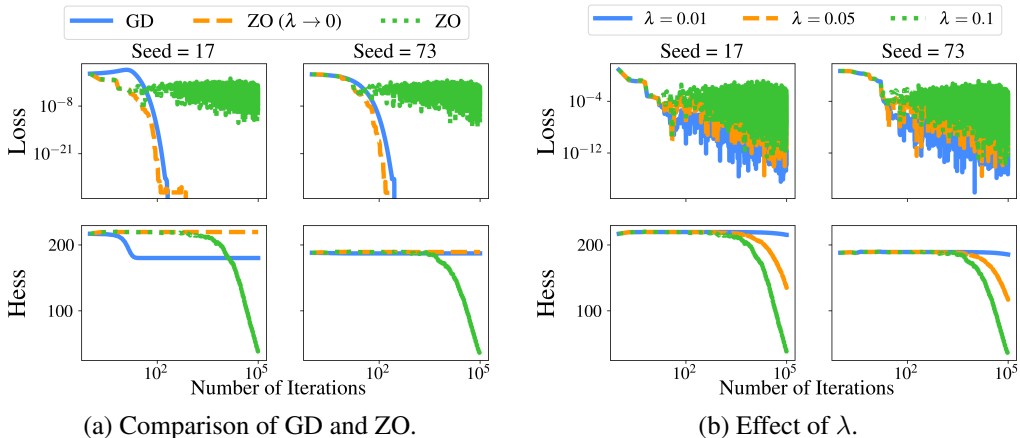

(a) Comparison of GD and ZO.

(b) Effect of $\lambda$.

Figure 4: Loss and trace of Hessian (Hess in the plot) on Example 2.1 using different random seeds. The observation aligns with Figure 1 (random seed 13). Zeroth-order optimization (ZO) decreases the trace of Hessian, and $\lambda$ controls the trade-offs between regularization on the trace of Hessian and the optimization error induced by additional bias terms.

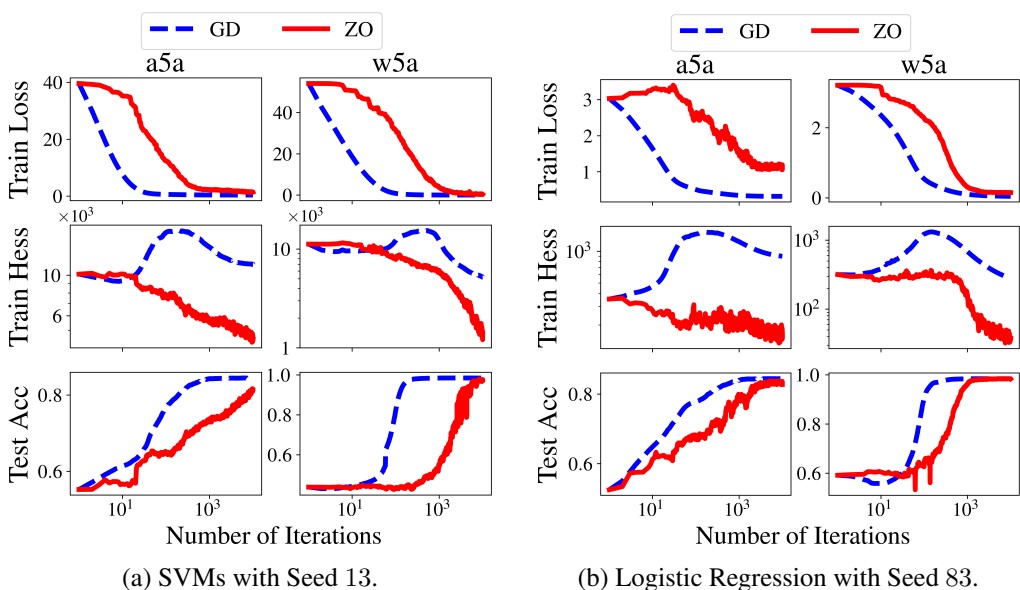

(a) SVMs with Seed 13.

(b) Logistic Regression with Seed 83.

Figure 5: Training loss, trace of Hessian on the training data (Train Hess in the plot), and test accuracy on (a) SVMs and (b) Logistic regression using different random seeds. The observation is consistent with Figure 2 (random seed 29), where zeroth-order optimization (ZO) reduces the trace of Hessian.

## D.2 Binary Classification with SVMs and Logistic Regression

The datasets "a5a" and "w5a" used in both the SVMs and logistic regression experiments are standard binary classification benchmarks from the LIBSVM library[1]. LIBSVM [14] is released under the BSD 3-Clause "New" or "Revised" License[2]. The a5a dataset contains $N = 6,414$ training and $26,147$ test samples with $d = 123$ features, and the w5a dataset contains $N = 9,888$ training and $39,861$ test samples with $d = 300$ features. We set $D = 10,000$ to overparameterize and run all algorithms for $T = 10,000$ iterations. To mitigate the effect of mini-batch noise, we use full-batch gradient descent and zeroth-order optimization in the experiments. The stepsize and smoothing

---

[1] https://www.csie.ntu.edu.tw/~cjlin/libsvmtools/datasets/binary.html
[2] https://github.com/cjlin1/libsvm/blob/master/COPYRIGHT

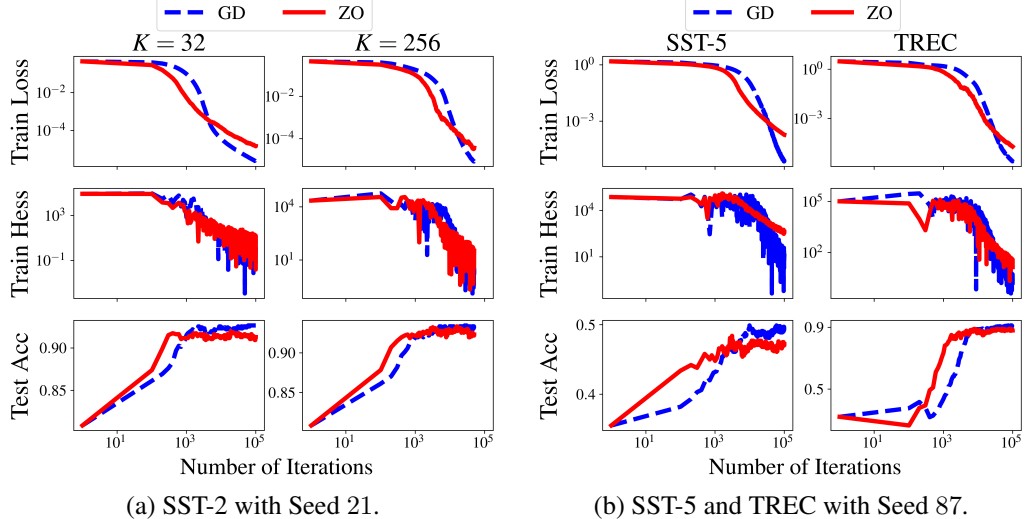

| (a) SST-2 with Seed 21. | (b) SST-5 and TREC with Seed 87. |

Figure 6: Training loss, trace of Hessian on the training data (Train Hess in the plot), and test accuracy on (a) SST-2 with $K = 32$ and $K = 256$, and (b) SST-5 and TREC with $K = 32$ using different random seeds. The observation is consistent with Figure 3 (random seed 42).

parameter $\lambda$ are searched from the grid $\{0.5, 0.1, 0.05, 0.01, 0.005, 0.001, 0.0005, 0.0001, 0.00005\}$. Gradient descent uses a stepsize of $0.001$ for SVMs and $0.01$ for logistic regression. For zeroth-order optimization, the selected hyperparameters are listed below.

- For SVMs, the stepsize is $0.0001$, and the smoothing parameter $\lambda = 0.05$.
- For logistic regression on a5a, the stepsize is $0.01$, and the smoothing parameter $\lambda = 0.1$.
- For logistic regression on w5a, the stepsize is $0.005$, and the smoothing parameter $\lambda = 0.05$.

The initialization is drawn from the Gaussian distribution $\mathcal{N}(0, (0.1)^2 \mathrm{I}_D)$. Randomness arises from the initialization, the random matrix $W$ used in $\phi(a_i) = W a_i$, and the random search directions in zeroth-order optimization. Figure 2 uses random seed 29, and Figure 5 reports results for additional seeds 13 and 83. The results are consistent across all 3 seeds. Each run is executed on a CPU and takes approximately 3.5 hours.

### D.3 Fine-Tuning Language Models on Text Classification Tasks

We follow experiment settings in Malladi et al. [69] and consider few-shot fine-tuning on RoBERTa-Large[3] (355M parameters) [61] with $K = 32$ and $K = 256$ examples per class on 3 text classification tasks: SST-2[4] and SST-5[5] [82] for sentiment classification, and TREC[6] [84] for topic classification. RoBERTa is available under the MIT License[7]. The datasets are commonly-used benchmarks that are publicly available for research purposes. Our implementation is based on the codebase provided by Malladi et al. [69] and uses the same prompts. Their code is released under the MIT License[8]. The SST-2, SST-5, and TREC datasets contain 2, 5, and 6 classes, respectively. We consider $K = 256$ only for the SST-2 dataset, and $K = 32$ for all 3 datasets. The training set is constructed by sampling $K$ examples per class from the original dataset, and the test set is built by randomly selecting $1,000$ examples from the original test dataset.

We fix the number of iterations to $100,000$ for $K = 32$ and $50,000$ for $K = 256$. The trace of Hessian is evaluated every 100 steps using the expected sharpness $(2/\delta^2)|\mathbb{E}_{u \sim \mathcal{N}(0, \mathrm{I}_d)}[f(x + \delta u)] - f(x)|$

---

[3]https://huggingface.co/FacebookAI/roberta-large
[4]https://huggingface.co/datasets/stanfordnlp/sst2
[5]https://nlp.stanford.edu/sentiment/
[6]https://huggingface.co/datasets/CogComp/trec
[7]https://github.com/facebookresearch/fairseq/blob/main/LICENSE
[8]https://github.com/princeton-nlp/MeZO/blob/main/LICENSE

Table 2: Total runtime (hours) and memory consumption (MiB) when fine-tuning RoBERTa using gradient descent (GD) and zeroth-order (ZO) optimization. We report both mean and standard error of the runtime across three random seeds $\{42, 21, 87\}$. Due to additional forward passes required to approximate the trace of Hessian, the reported runtime is longer than standard training. GD consumes less memory than the default first-order optimizer, AdamW, which stores additional optimizer states.

| Task | | SST-2 | | SST-5 | TREC |
| --- | --- | --- | --- | --- | --- |
| | | $K = 256$ | $K = 32$ | $K = 32$ | |
| Batch Size | | 512 | 64 | 160 | 192 |
| Number of Iterations | | $50,000$ | $100,000$ | $100,000$ | $100,000$ |
| Runtime | GD | $29.47 \pm 0.93$ | $7.38 \pm 0.33$ | $21.64 \pm 1.18$ | $14.79 \pm 0.56$ |
| | ZO | $25.13 \pm 0.94$ | $7.34 \pm 0.22$ | $18.91 \pm 0.52$ | $13.37 \pm 0.47$ |
| Memory | GD | $54,214$ | $7,986$ | $18,826$ | $14,258$ |
| | ZO | $5,038$ | $3,074$ | $3,440$ | $3,274$ |

as an approximation, where $\delta = 10^{-4}$ and the expectation is estimated by averaging over 100 samples. To reduce the impact of mini-batch noise, we use full-batch gradient descent and zeroth-order optimization in all experiments, both without learning rate scheduling. This leads to batch sizes of 512 (SST-2 with $K = 256$), 64 (SST-2 with $K = 32$), 160 (SST-5 with $K = 32$), and 192 (TREC with $K = 32$). The stepsize for gradient descent is set to $5 \times 10^{-5}$ for all cases. Zeroth-order optimization uses a stepsize of $10^{-5}$ for SST-2, and $5 \times 10^{-6}$ for SST-5 and TREC. The smoothing parameter $\lambda$ in zeroth-order optimization is set to $2 \times 10^{-3}$ for SST-2 and TREC, and $10^{-3}$ for SST-5. Randomness arises from the selection of datasets, the initialization, and the random search directions in zeroth-order optimization. Figure 3 uses random seed 42, and Figure 6 reports results for additional seeds 21 and 87. The results are consistent across all 3 seeds. All experiments are tested on a single NVIDIA H100 GPU with 80 GiB memory. The runtime and memory are summarized in Table 2.

