# OpenReview forum: "Zeroth-Order Optimization Finds Flat Minima"
_NeurIPS.cc/2025/Conference — NeurIPS 2025 poster_

### Official Review · Reviewer_79de · 2025-06-05

**Clarity:** 3
**Significance:** 1
**Originality:** 3
**Rating:** 4
**Confidence:** 3

**Summary:**

This paper investigates the implicit regularization properties of zeroth-order (ZO) optimization methods, specifically focusing on the behavior of the standard two-point gradient estimator. The central claim of the paper is that ZO optimization with the two-point estimator implicitly favors and converges to "flat minima". The authors provide the theoretical analysis to support their method. Also, authors provide empirical validation across several settings to support their theoretical findings.

**Questions:**

See the weaknesses part.

**Ethical Concerns:**

["NO or VERY MINOR ethics concerns only"]

**Final Justification:**

My concerns have been addressed.

**Limitations:**

Yes.

**Paper Formatting Concerns:**

No formatting issues.

**Quality:**

2

**Strengths And Weaknesses:**

Strengths:
1.  Rigorous Theoretical Analysis for Convex Case.
2. The paper is generally well-written and logically structured.
3. The experimental results provide support for the theoretical claims, especially in convex settings.

Weaknesses:
1. Previous work on SAM-based models has primarily focused on analyzing their methods in non-convex cases. If the entire region under consideration were convex and smooth, it is likely that many existing methods would demonstrate good properties. Therefore, evaluating the theoretical findings of this paper becomes challenging. I believe the lack of analysis for non-convex cases and the absence of a theoretical comparison with other methods weaken the paper's contribution.

2. It is surprising that a two-point gradient estimator can achieve good performance in learning tasks. In my opinion, this type of method lacks second-order information and should theoretically struggle or fail in ill-conditioned optimization problems. The authors should provide numerical results on well-known optimization problems, such as the Lévy, Ackley, and Rastrigin functions, to support their method's convergence claims.

3. The paper lacks a sufficient comparison with baseline methods. I believe there are more than ten relevant zeroth-order (and potentially simple first-order) methods, but the authors choose not to compare their approach against any of them.

4. In Figure 3, it appears that the methods have not fully converged.

---

> ### Author Rebuttal · Authors · 2025-07-30
>
> Many thanks for the detailed review and instructive feedback. We provide clarifications below.
>
> 1. **Challenging to evaluate due to smoothness and convexity assumptions.**
>
> > Previous work on SAM-based models has primarily focused on analyzing their methods in non-convex cases. If the entire region under consideration were convex and smooth, it is likely that many existing methods would demonstrate good properties. Therefore, evaluating the theoretical findings of this paper becomes challenging. I believe the lack of analysis for non-convex cases and the absence of a theoretical comparison with other methods weaken the paper's contribution.
>
> Thanks for this valuable comment. We agree that convergence analysis on nonconvex functions will make our paper even more solid, but we respectfully disagree that only considering smooth and convex functions directly suggests that our results are not significant and challenging to evaluate.
>
> * **Significance of our results.** First, the implicit regularization of zeroth-order optimization in Eq. (2) and (3) **does not require** the smoothness and convexity assumptions and holds for **any** twice continuously differentiable functions. The test function in Figure 1 is nonsmooth and nonconvex, but zeroth-order optimization is capable of decreasing both the objective value and the trace of Hessian.
> Second, we provide the first convergence guarantees towards flat **global** minima. This can be extended to the case where the function is only **locally convex** in a neighborhood; see Remark 3.2.
>
> * **Existing theoretical understanding of SAM-based methods.** The work [11] mainly focused on convex quadratic functions that are **smooth and convex**. They also discussed nonconvex functions that are **locally quadratic** and provided implicit regularization of SAM in the form of Eq. (3).
> [89] did not assume convexity and considered local convergence behaviors for **smooth** functions. They instead assumed existence of a low-dimensional solution manifold $\Gamma$ consisting of local minimizers with low-rank Hessian matrix and supposed that gradient flow admits attraction set $U$ on this solution manifold $\Gamma$. When **initializing on the attraction set** $U$, SAM converges to flat minima with the smallest trace of Hessian on the set $U$.
> [3] considered **smooth functions locally satisfying the PL condition**. Convergence to **stationary points** of the trace of Hessian
> evaluated at limit points under gradient flow was established.
> Compared with these methods, $(i)$ we adopt a global definition of flat minima that are more challenging to guarantee; $(ii)$ it is not immediate to conclude that they have weaker assumptions of the objective functions. Therefore, we do not think the fact that these paper studied nonconvex functions weakens our contributions.
>
> * **More on smooth and convex functions.** It is difficult (and sometimes meaningless) to derive algorithms and theoretical complexities with entire generality, such as problems with arbitrary nonconvexity. That's exactly why the majority of literature associates complexity bounds with problem classes. Smooth and convex functions, as an example, are still general enough to **capture interesting problems** and **provide new insights** to guide both the design and analysis of algorithms for general nonconvex functions.
> Logistic regression, as an example of smooth and convex functions, remains an important problem in learning theory and continually reveals new open directions; see (COLT, 2019) and (ICML, 2025) below.
> Nesterov momentum [71] achieves accelerated convergence rates only on smooth and convex functions, but this does not affect its significance for understanding momentum-based methods central to the success of modern deep learning.
> Among the most significant optimization techniques of the last decade, variance reduction methods like SAGA and SVRG were first developed only for smooth and convex objectives.
>
> We consider smooth and convex functions as an initial step to analyze convergence to flat minima for zeroth-order optimization. We believe our results provide new insights in the understanding of zeroth-order optimization and open up promising directions in the study of nonconvex functions.
>
> **References**
>
> (COLT, 2019) The Implicit Bias of Gradient Descent on Nonseparable Data.
>
> (ICML, 2025) Benefits of Early Stopping in Gradient Descent for Overparameterized Logistic Regression.
>
>
> 2. **It is surprising that ZO achieves good performance.**
>
> > It is surprising that a two-point gradient estimator can achieve good performance in learning tasks. In my opinion, this type of method lacks second-order information and should theoretically struggle or fail in ill-conditioned optimization problems. The authors should provide numerical results on well-known optimization problems, such as the Lévy, Ackley, and Rastrigin functions, to support their method's convergence claims.
>
> The two-point estimator is one of the most popular and standard methods used in zeroth-order optimization. There are **extensive evidences** that zeroth-order optimization can achieve good performance in learning tasks. We refer the reviewer to the work [74] and subsequent literature citing it. Here, we only mention LLMs fine-tuning as a challenging nonconvex learning task. [69] has already reported that zeroth-order optimization is capable of achieving similar performance as first-order methods on many LLMs fine-tuning tasks. We also provide examples in Figures 1 (nonconvex and nonsmooth test function), 2 (convex learning tasks), and 3 (LLMs fine-tuning) demonstrating that zeroth-order optimization achieves good performance.
>
> Although lacking **explicit** second-order information, Eq. (2) says that zeroth-order optimization can incorporate trace of Hessian **implicitly** in its objective function, which is the major intuition motivating this work.
> We agree that explicit second-order information provides richer information of the loss landscape and that second-order methods converge faster on many objective functions. However, second-order information can be computationally expensive when scaling to large model training.
> The optimizers behind the success of modern deep learning, such as SGD, SAM, and AdamW, are all first-order methods lacking explicit second-order information. They are powerful enough to achieve nice performance.
>
> Following the reviewer's suggestion, we examine the behaviors of zeroth-order optimization on the Levy, Ackley, and Rastrigin functions. Note that all these functions **only have one global minimizer**, and local minimizers are **isolated** from each other, unlike the nonconvex test function $(y^\top z-1)^2/2$ in Figure 1 where all points satisfying $y^\top z=1$ are global minima and form a **connected** solution manifold.
> On the Levy, Ackley, and Rastrigin functions, behaviors of algorithms crucially depend on the initialization rather than implicit regularization.
> On the test function in Figure 1, there are infinitely many global minimizers, and implicit regularization selects which minima to converge to.
> Therefore, the Levy, Ackley, and Rastrigin functions are not good examples to test implicit regularization, and we only report loss values to show that zeroth-order optimization is capable of minimizing them. We choose the dimension to be 100 for all three functions and run different algorithms for $10^4$ steps.
> "No. of Steps To Converge" is the iteration where the loss stabilizes and the absolute difference between the current loss and the final loss is less than $10^{-15}$.
>
> | Function | Initial Loss |  | Final Loss | | No. of Steps To Converge | |
> | -- | -- | -- | -- | -- | -- | -- |
> | | ZO | GD | ZO | GD | ZO | GD |
> | Levy | 0.622 | 0.622 | 8e-4 | 0 | Not Converged at 10,000 | 7605 |
> | | 12.39 | 12.39 | 0.903 | 0.991 | Not Converged at 10,000 | 8085 |
> | Ackley | 0.967 | 0.967 | 4e-15 | 8e-3 | 9167 | 2333 |
> | | 3.958 | 3.958 | 1.605 | 1.646 | Not Converged at 10,000 | 2393 |
> | Rastrigin | 214.29 | 214.29 | 0 | 0 | 5578 | 37 |
> |  | 1147.7 | 1147.7 | 36.81 | 35.82 | 6537 | 54 |
>
>
> 3. **Compare with baseline methods.**
>
> > The paper lacks a sufficient comparison with baseline methods. I believe there are more than ten relevant zeroth-order (and potentially simple first-order) methods, but the authors choose not to compare their approach against any of them.
>
> It seems there might be a misunderstanding.
> Our paper mainly studies implicit regularization of zeroth-order optimization with the standard two-point estimator.
> Evaluating the implicit regularization of additional benchmarks, as well as faster-converging algorithms, lies beyond the scope of this work as we make no claims about them.
> The goal of our empirical experiments is only to evaluate whether zeroth-order optimization with the two-point estimator decreases the trace of Hessian.
> In this regard, we believe our current experiment results suffice to support the theoretical findings.
>
>
> 4. **Not fully converge on LLMs.**
>
> > In Figure 3, it appears that the methods have not fully converged.
>
> LLMs fine-tuning is a challenging task. Depending on the dataset, it is often not possible or takes extremely long time to achieve full convergence. The runtime of Figure 3 can be found in Table 2 in the appendix. The current results already take 7 - 30 hours for a single run.
> Even though the loss might continue to decrease with more iterations, the trend of decreasing trace of Hessian for zeroth-order optimization is already clear and thus suffices to support our claims.

---

> > ### Comment · Reviewer_79de · 2025-08-04
> >
> > Thanks for your reply. After reading the other reviewers' comments and the authors' responses, I believe this is an interesting research paper and I have increased my score. Some papers also discuss finding flat minima or the sharpness of optimization with only forward passes (e.g., "SharpZO: Hybrid Sharpness-Aware Vision Language Model Prompt Tuning via Forward-Only Passes" and "Sharpness-Aware Black-Box Optimization"). The authors should add a discussion on the differences between these papers and their work. I now understand that all experiments in this paper are intended only to verify that ZO can reach flat minima. Therefore, I think the authors should provide multiple runs (with different seeds) and multiple settings (with different learning rates) to enhance the results. Additionally, it is quite unfair to plot GD and ZO on the same graph, as they use different step sizes. GD could converge very fast, while two-point ZO naturally converges slower.

---

> > > ### Author Response · Authors · 2025-08-06
> > >
> > > Thank you for taking the time to read our response, as well as the other reviewers’ comments and corresponding replies. We sincerely appreciate your re-evaluation and that you find the paper interesting.
> > >
> > > We will add the discussions to include these relevant papers in the revision. We have provided some experiments using different random seeds in the appendix, and the results are consistent. We will add more experiments in the revision to provide multiple runs and multiple settings using different learning rates. Many thanks for the additional suggestions to improve our work!

---

### Official Review · Reviewer_erZq · 2025-06-24

**Clarity:** 4
**Significance:** 3
**Originality:** 4
**Rating:** 4
**Confidence:** 4

**Summary:**

The paper shows that a smoothed function based on a two-point zeroth-order gradient estimator incorporates the trace of the Hessian matrix implicitly as a regularization term. Zeroth-order minimization thus implicitly favors small eigenvalues of the Hessian. A small trace of the Hessian matrix has been shown to correlate with flat minima. The authors give a proof of convergence of zeroth-order optimization to flat minima, and show experimentally that the trace of the Hessian is smaller for zeroth-order than for first-order optimization methods.

**Questions:**

1. Zeroth-order optimization is particularly interesting if it enables optimization of non-differentiable functions by Gaussian smoothing. However, your work depends on twice continuously differentiable objective functions (eq. 2). Do you have an idea if your work could be extended to non-smooth C^{0,0} functions?
2. Why is it that experimental results consistent with the theory are only found for convex losses (SVM, logistic regression), but not for non-convex losses (LLMs)? Is the convexity assumption perhaps more that a theoretical tool to prove convergence?
3. Isn't the main claim of convergence to flat minima the fact that such methods show improved generalization? Is this the case in your experiments?

**Ethical Concerns:**

["NO or VERY MINOR ethics concerns only"]

**Limitations:**

yes

**Paper Formatting Concerns:**

none found

**Quality:**

3

**Strengths And Weaknesses:**

Strengths:
- Nice use of Taylor's theorem to show the connection of two-point gradient approximation and Hessian matrix (eq.2)
- Rigorous mathematical proof of convergence

Weaknesses:
- Very strong assumptions of function to be optimized being twice continuously differentiable and convex
- Experimental results consistent with the theory are only found for convex losses (SVM, logistic regression), but not for non-convex losses (LLMs)
- No attempt to link flat minima to improved generalization

---

> ### Author Rebuttal · Authors · 2025-07-30
>
> We thank the reviewer for the insightful questions. Here are our detailed clarifications.
>
> 1. **Strong assumptions.**
>
> > Very strong assumptions of function to be optimized being twice continuously differentiable and convex. Zeroth-order optimization is particularly interesting if it enables optimization of non-differentiable functions by Gaussian smoothing. However, your work depends on twice continuously differentiable objective functions (eq.2). Do you have an idea if your work could be extended to non-smooth $C^{0,0}$ functions?
>
> Zeroth-order optimization is previously popular for optimizing nonsmooth functions where gradients do not always exist.
> However, even when gradients can be computed, there are scenarios where zeroth-order optimization **still offer clear advantage** over gradient-based methods in terms of memory consumption, i.e., LLMs fine-tuning tasks.
> For example, [69] reported that 8 A100 GPUs, each with 80 GiB of memory, are required for gradient-based methods to fine-tune a 30-billion-parameter model, whereas a single GPU is enough for zeroth-order optimization.
> Therefore, we focus on the case when the function is differentiable. Note that our current definition of flat minima requires Hessian information, and thus we assume the function to be twice differentiable such that the trace of Hessian is well-defined.
> Extension to nonsmooth functions requires a different measure of sharpness. A promising alternative is to define flat minima as minima with the smallest $f_\lambda(x) - f(x)$. Detailed study is left for future work.
>
> We consider smooth and convex functions as an initial study of convergence to flat **global** minima for zeroth-order optimization. We want to emphasize that such guarantees remain unknown before our work in the existing literature.
> We believe our results open up promising directions in the study of flat minima, including a better understanding for nonsmooth and nonconvex functions.
>
>
> 2. **Experiments seem not consistent.**
>
> > Experimental results consistent with the theory are only found for convex losses (SVM, logistic regression), but not for non-convex losses (LLMs). Is the convexity assumption perhaps more that a theoretical tool to prove convergence?
>
> Our theoretical results only consider zeroth-order optimization, with implicit regularization towards small trace of Hessian for **any** twice continuously differentiable function in Eq. (2) and (3), and convergence rates to flat minima for convex functions in Corollary 2. We do not argue and claim any behaviors of first-order optimization for general functions.
> For LLMs fine-tuning tasks, it can be observed that the estimated trace of Hessian decreases for zeroth-order optimization. Therefore, the experiment results are **also consistent**.
> Note that the test function in Figure 1 is also **nonconvex**, and the trace of Hessian also decreases for zeroth-order optimization. This aligns with the implicit regularization in Eq. (2) and (3) that **does not require the convexity assumption**.
>
> The reason why GD also decreases the trace of Hessian in some settings is that GD also has implicit regularization. For example, GD with large learning rates penalizes the largest eigenvalue of Hessian [20, 7]; Reviewer 2MGS also points out a recent paper showing that full-batch GD converges to flat minima in shallow linear networks.
> The study of implicit regularization of first-order optimization is not the primary focus of our paper. It still remains an interesting question when and why such implicit regularization happens. **We believe this depends crucially on the loss landscape, the initialization point, and the learning rate**. A more detailed study on this topic will be valuable to the community.
>
>
> 3. **Flat minima and generalization.**
>
> > No attempt to link flat minima to improved generalization. Isn't the main claim of convergence to flat minima the fact that such methods show improved generalization? Is this the case in your experiments?
>
> We explicitly avoid claiming that zeroth-order optimization generalizes better as it finds flat minima.
>
> * The link between flat minima and improved generalization is still an ongoing research direction. Although there are evidences that flat minima generalize better and methods designed for finding flat minima have better empirical performance, there are also cases where sharp minima can generalize [25] and flat minima do not generalize [88].
> In our experiments, we find that GD often converges to flatter minima and has better test accuracy on LLMs fine-tuning tasks, but it is not the case for binary classification tasks.
>
> * The measure of sharpness and definition of flat minima are also undergoing research in the community. In addition to the trace of Hessian used in this work, there are also other widely-adopted notions such as the largest eigenvalue of the Hessian. There are still no agreement on which is the most precise measure that corresponds to generalization performance.
>
> Given these arguments, we only claim in the paper that zeroth-order optimization has implicit regularization towards flat minima, where we adopt the trace of Hessian to measure sharpness. Such implicit regularization is an interesting algorithmic property to explore, as in first-order methods [103, 12, 11], and improves our understanding of zeroth-order optimization. The link between flat minima and improved generalization is not the main focus of the work.

---

### Official Review · Reviewer_9wKN · 2025-06-24

**Clarity:** 3
**Significance:** 3
**Originality:** 3
**Rating:** 4
**Confidence:** 2

**Summary:**

This paper studies the convergence of zeroth-order algorithms toward flat minima. Specifically, the authors analyze the convergence rate of a zeroth-order algorithm employing the standard two-point estimator, to solutions with minimal Hessian trace within the optimal solution set for smooth convex functions.
 Experiments on SVM, logistic regression, and language model fine-tuning validate the theoretical findings.

**Questions:**

1. Can the results in this paper be extended to non-convex settings? Since the derivation of implicit regularization (Eq.(3)) does not require convexity assumptions.

2. In Figure 3, it seems that GD is more effective than ZO in reducing the Hessian trace. Could the authors provide further explanation for this phenomenon?

**Ethical Concerns:**

["NO or VERY MINOR ethics concerns only"]

**Final Justification:**

For the extension to the nonconvex setting, the authors have outlined the current challenges as well as potential solutions. Regarding the experimental behavior of GD, the authors also provided references that offer explanations. Since these issues are not the main focus of this work, and given that this paper is the first to study the implicit regularization of sharpness by zeroth-order algorithms, I recommend accepting this paper.

**Limitations:**

Yes

**Quality:**

3

**Strengths And Weaknesses:**

Strengths

1. This paper is well written and easy to follow. The derivation of implicit regularization and flat minima (Sections 2 and 3) are clear and easy to understand.

2. The derivation of implicit regularization on Hessian trace in zeroth-order algorithms is both novel and theoretically interesting.

3. The convergence analysis in Section 3.1 is clearly presented, offering valuable theoretical insights.


Weaknesses

1.  This paper primarily focuses on the convex setting, where the Hessian matrix is positive semi-definite, making the trace of the Hessian an appropriate measure of sharpness. However, it remains unclear whether the conclusions in this paper hold in non-convex scenarios, although the derivation of implicit regularization on the Hessian trace does not require convexity assumptions.

---

> ### Author Rebuttal · Authors · 2025-07-30
>
> We appreciate the reviewer for highlighting the strengths of our work. We address additional concerns in the following.
>
> 1. **Extension to nonconvex functions.**
>
> > It remains unclear whether the conclusions in this paper hold in non-convex scenarios, although the derivation of implicit regularization on the Hessian trace does not require convexity assumptions. Can the results in this paper be extended to non-convex settings?
>
> Although the implicit regularization in Eq. (3) also holds for nonconvex functions, the precise convergence analysis to flat minima requires convexity assumption in the current work.
> We briefly discuss extensions to the nonconvex setting in Remark 3.2, where the current results can be directly extended to the case where the function is only locally convex in a neighborhood.
> For general nonconvex functions, convergence analysis to flat minima requires both different definitions of flat minima and novel analysis techniques with additional assumptions. A detailed study is left for future.
> Nonetheless, we want to emphasize that convergence guarantees to our definition of flat **global** minima remain unknown before this work in the existing literature.
> We believe our results provide new insights and present a first step towards more challenging settings, including nonconvex problems.
>
>
> 2. **Explanation for GD.**
>
> > In Figure 3, it seems that GD is more effective than ZO in reducing the Hessian trace. Could the authors provide further explanation for this phenomenon?
>
> GD also has implicit regularization. For example, GD with large learning rates penalizes the largest eigenvalue of Hessian [20, 7]; Reviewer 2MGS also points out a recent paper showing that full-batch GD converges to flat minima in shallow linear networks.
> The study of implicit regularization of first-order optimization is not the primary focus of our paper. It still remains an interesting question when and why such implicit regularization happens. **We believe this depends crucially on the loss landscape, the initialization point, and the learning rate**.

---

> > ### Comment · Reviewer_9wKN · 2025-08-03
> >
> > I thank authors for providing detailed replies for all my concerns. I maintain my current score to recommend acceptance.

---

### Official Review · Reviewer_2MGS · 2025-06-30

**Clarity:** 3
**Significance:** 3
**Originality:** 3
**Rating:** 5
**Confidence:** 4

**Summary:**

This paper demonstrates an implicit bias in two-point zeroth-order optimization of $C^3$ functions towards flatter minima, as measured by the Hessian trace, and provides convergence analysis. This is done via a smoothed surrogate, which is shown to implicitly encode a regularizer term containing the Hessian trace. The smoothing parameter is framed as a trade-off between this regularization and the optimization error induced by a bias term. Experiments on binary classification problems in the overparameterized setting, and fine-tuning language models are shown.

**Questions:**

1. Could you clarify the $2/\delta^2$ constant in the sharpness estimation equation, via a simple derivation, for instance?
2. Do the results of your experiments change when the grid search is used to prioritize both validation loss and Hessian norm? If not, can you comment on why your experiments show such a strong bias (as a trade-off for convergence rate) for low Hessian trace compared to batch GD?

**Ethical Concerns:**

["NO or VERY MINOR ethics concerns only"]

**Final Justification:**

All of my concerns have been addressed, and the paper makes interesting and novel contributions to the field. I recommend acceptance.

**Limitations:**

yes

**Quality:**

3

**Strengths And Weaknesses:**

## Strengths

* This paper has very clear writing and proofs are relatively easy to follow. The paper also clearly distinguishes its analysis from that provided by Nesterov.
* The paper addresses an important problem of explicitly connecting zeroth-order optimization to the research on flat minima.

## Weaknesses

* In Appendix D.2, the authors state that they chose the learning rate and smoothing parameter via grid search. What metric was used to perform this? In [1], the authors, show the batch GD (which is used in this paper) can also converge to flatter minima (compared to gradient flow), and establish a linear convergence rate under the edge of stability regime. Therefore, this paper's evidence may be stronger if the metric used to compare also takes into account the sharpness (via the Hessian norm, for example) of the solutions, rather than just a validation loss (which I assume was used here).
* On L297 (page 9), can you expand on the $2/\delta^2$ constant in your estimation of sharpness? I could not find that leading constant in either Neyshabur et al. (ref. 75) or Zhu et al. (ref. 103). In the former, the closest equation is (5), which uses $\mathbb{E}\_{u \sim \mathcal{N}(0, \sigma I)} [f(x + u)] - f(x)$, and Zhu et al. use (from their Supplementary), $\mathbb{E}_{u \sim \mathcal{N}(0, \delta^2 I)} [f(x + u)] - f(x)$.
* There are relatively few experiments, and these do not account for the stochastic nature of the methods (caused, e.g., by random initialization). That said, I do not think empirical analysis is the point of this paper, and that the experiments are moreso for a cursory confirmation of the established theory.

[1] Beneventano, P., & Woodworth, B. (2025). Gradient Descent Converges Linearly to Flatter Minima than Gradient Flow in Shallow Linear Networks. arXiv preprint arXiv:2501.09137.

---

> ### Author Rebuttal · Authors · 2025-07-30
>
> We thank the reviewer for the positive feedback and insightful questions. Here are our clarifications.
>
> 1. **Metric used for model selection.**
>
> > The authors state that they chose the learning rate and smoothing parameter via grid search. What metric was used to perform this? This paper's evidence may be stronger if the metric used to compare also takes into account the sharpness (via the Hessian norm, for example) of the solutions, rather than just a validation loss (which I assume was used here). Do the results of your experiments change when the grid search is used to prioritize both validation loss and Hessian norm?
>
> We follow standard procedure in machine learning and only use the accuracy as the metric to select the best learning rate and smoothing parameter. We do not directly take into account the sharpness as metric since
>
> * This introduces bias into model selection and affects our evaluation of implicit regularization, since parameters that lead to decreased trace of Hessian are always selected.
>
> * The link between sharpness and model performance is an ongoing research direction. The claim that sharp minima always lead to worse model performance is still debatable [25]. Therefore, we directly use model performance as the metric.
>
> Although the experiment results depend on the hyperparameters, we find that the trend of decreasing trace of Hessian is **consistent** across choices of learning rates and smoothing parameters such that the two-point estimator results in good model performance.
>
>
> 2. **Compare to batch GD.**
>
> > In [1], the authors show the batch GD (which is used in this paper) can also converge to flatter minima (compared to gradient flow) and establish a linear convergence rate under the edge of stability regime. Can you comment on why your experiments show such a strong bias (as a trade-off for convergence rate) for low Hessian trace compared to batch GD?
>
> Thanks for bringing this interesting paper into our attention. We will include it in the reference of our paper.
> The study of implicit regularization of first-order optimization is not the primary focus of our paper. It still remains an interesting question when and why such implicit regularization happens.
> In our experiments, although the behavior of zeroth-order optimization is consistent across all three settings we consider, the behavior of GD varies; the trace of Hessian decreases for LLMs fine-tuning tasks but not for the test function and the binary classification tasks.
> **We believe this depends crucially on the loss landscape, the initialization point, and the learning rate**.
> We leave a detailed study to future work.
>
>
> 3. **Constant in the expected sharpness.**
>
> > Could you clarify the $2/\delta^2$ constant in the sharpness estimation equation, via a simple derivation, for instance?
>
> This follows from similar reasons as Eq.(2):
> \begin{equation*}
>     \mathbb{E}_{u\sim \mathcal{N}(0, \delta^2 I)}[f(x+u)] - f(x)
>     =
>     f\_\delta(x) - f(x)
>     =
>     \frac{\delta^2}{2} \text{Tr}(\nabla^2 f(x)) + o(\delta^2).
> \end{equation*}
> Therefore, we add this $2/\delta^2$ constant to directly estimate the trace of Hessian.
>
>
> 4. **Relatively few experiments.**
>
> > There are relatively few experiments, and these do not account for the stochastic nature of the methods (caused, e.g., by random initialization). That said, I do not think empirical analysis is the point of this paper, and that the experiments are more for a cursory confirmation of the established theory.
>
> Thanks for this comment. In Appendix D, we provide more experiments using different random seeds to measure the effect of random initialization and random search directions in zeroth-order optimization. The results are consistent in all our settings. The main focus of the paper is on the theoretical properties of zeroth-order optimization, and current experiments suffice to act as confirmation of our theory. We leave a more extensive empirical study to future work.

---

> ### Comment · Reviewer_2MGS · 2025-08-04
>
> Thank you for your rebuttal. My primary concerns have been addressed, and I maintain my original recommendation of acceptance.

---

### Official Review · Reviewer_WS1L · 2025-07-02

**Clarity:** 3
**Significance:** 2
**Originality:** 3
**Rating:** 4
**Confidence:** 3

**Summary:**

This paper presents a theoretical analysis of zeroth-order (ZO) optimization, revealing its implicit regularization effect towards finding flat minima.

The authors provide a theoretical framework to explain this phenomenon
The core idea—that the smoothing inherent in ZO methods acts as a regularizer minimizing the trace of the Hessian—is both insightful and elegant.

The theoretical contributions are well-supported by proofs.
While the experiments provide corroborating evidence, there are some questions regarding the practical implications in highly non-convex settings.
Overall, this is a paper that makes contribution to our understanding of zeroth-order methods

**Questions:**

This raises the question of the practical utility of using ZO for this task.

If ZO does not offer a clear advantage in either final performance or the "flatness" of the solution compared to a well-tuned first-order method, what is the key takeaway for a practitioner?
The paper convincingly shows that ZO also seeks flat minima, but it does not make a strong case for why one would prefer it over GD in this realistic, non-convex scenario.

**Ethical Concerns:**

["NO or VERY MINOR ethics concerns only"]

**Quality:**

3

**Strengths And Weaknesses:**

Strengths:

Novelty and Conceptual Contribution: The primary strength of this work lies in its novel perspective. While the analysis of ZO methods via smoothed functions is standard, this paper is the first to re-interpret the Hessian trace term not as a bias to be suppressed, but as a desirable regularization term to be embraced. This leap, connecting a classic optimization technique to the modern pursuit of flat minima for better generalization, is a fundamental contribution.

Theoretical Rigor: The paper's claims are backed by solid theoretical work. The authors formally define flat minima in a global context, construct a novel proof strategy around a regularized objective function F(x), and provide a full convergence complexity analysis (T = O(d⁴/ε²)) for finding approximate flat minima. The technical depth required for the proofs, such as handling 8th-order moments of Gaussian variables, demonstrates the non-triviality of the theoretical analysis.

Points for improvement

While the theoretical contributions are clear, I have some questions regarding the experimental section, particularly concerning the practical implications of the findings in the non-convex LLM fine-tuning setting.

On the Practical Significance in the Non-Convex Setting: The authors are commended for extending their investigation to the challenging setting of fine-tuning a 355M parameter RoBERTa model. However, the practical significance of the findings is not entirely clear from the results presented. As shown in Figure 2 and 3, standard gradient descent (GD) often achieves a lower training loss, a smaller Hessian trace (i.e., finds an even "flatter" solution), and higher test accuracy.

---

> ### Author Rebuttal · Authors · 2025-07-30
>
> We thank the reviewer for acknowledging the contribution of our work. We address the reviewer's concern regarding practical implications in the following.
>
> 1. **Practical significance.**
>
> > The practical significance of the findings is not entirely clear from the results presented. As shown in Figure 2 and 3, standard gradient descent (GD) often achieves a lower training loss, a smaller Hessian trace (i.e., finds an even "flatter" solution), and higher test accuracy.
>
> Thanks for raising this point. The aim of the experiments is not to claim that zeroth-order optimization is better than first-order optimization, but only to provide examples to support our theoretical findings that the trace of Hessian decreases.
> In terms of training loss and test accuracy, there is still a gap between the performance of zeroth and first-order optimization, which is known in the literature [69, 100] and is supported by classical theoretical understandings that the convergence rates of zeroth-order optimization depend on the dimension.
> For the trace of Hessian, we provide the first evidence that zeroth-order optimization has implicit regularization to reduce the trace of Hessian.
> Despite this gap in performance, there are scenarios where first-order optimization cannot be applied, and we **have to** use zeroth-order optimization instead.
> For example, LLMs fine-tuning tasks where gradients are expensive to compute [69] and black-box settings where gradients are not accessible [17].
> We provide more clarification in the following reply.
>
>
> 2. **When to use ZO.**
>
> > If ZO does not offer a clear advantage in either final performance or the "flatness" of the solution compared to a well-tuned first-order method, what is the key takeaway for a practitioner? The paper convincingly shows that ZO also seeks flat minima, but it does not make a strong case for why one would prefer it over GD in this realistic, non-convex scenario.
>
> Zeroth-order optimization is often preferred over first-order optimization when gradients are not accessible or expensive to compute.
> In LLMs fine-tuning tasks, first-order optimization achieves better performance but has significantly larger memory consumption.
> For example, [69] reported that first-order optimization requires 8 A100 GPUs to fine-tune a 30-billion-parameter model, each with 80 GiB memory, whereas a single GPU is enough for zeroth-order optimization.
> In the case when resources are not sufficient, e.g., only one GPU is available, zeroth-order optimization still provides a promising alternative and democratizes the use of LLMs to users with **limited resources**.
> For practitioners, the major advantage of zeroth-order optimization is still on its memory efficiency, but we provide a new theoretical understanding on its implicit regularization towards flat minima, which is previously observed only for some first-order methods such as SAM [89, 11].

---

### Decision · Program_Chairs · 2025-09-17

**Decision:**

Accept (poster)

**Comment:**

This paper studies the implicit regularization of zeroth-order optimization with the two-point estimator, showing that it favors flat minima characterized by small Hessian trace. The authors also provide convergence rates in convex settings and validate the theory with experiments.

All reviewers are positive and agree that this paper is novel and interesting. The theoretical guarantees and supporting experiments are convincing, and initial concerns have been addressed in the rebuttal. I recommend acceptance. However, the authors are encouraged to further refine the final version by clarifying differences with related work and ensuring fair comparisons in the experiments.